# Adaptive Proteomic Changes in Protein Metabolism and Mitochondrial Alterations Associated with Resistance to Trastuzumab and Pertuzumab Therapy in HER2-Positive Breast Cancer

**DOI:** 10.3390/ijms26041559

**Published:** 2025-02-12

**Authors:** Juan Madoz-Gúrpide, Juana Serrano-López, Marta Sanz-Álvarez, Miriam Morales-Gallego, Socorro María Rodríguez-Pinilla, Ana Rovira, Joan Albanell, Federico Rojo

**Affiliations:** 1Department of Pathology, Fundación Jiménez Díaz University Hospital Health Research Institute (IIS—FJD, UAM)—CIBERONC, 28040 Madrid, Spainmiriam.moralesg@quironsalud.es (M.M.-G.); smrodriguez@quironsalud.es (S.M.R.-P.); 2Department of Haematology, Fundación Jiménez Díaz University Hospital Health Research Institute (IIS—FJD, UAM)—CIBERONC, 28040 Madrid, Spain; juana.serrano@quironsalud.es; 3Cancer Research Program, IMIM (Hospital del Mar Research Institute), 08003 Barcelona, Spain; arovira@imim.es; 4Department of Medical Oncology, Hospital del Mar—CIBERONC, 08003 Barcelona, Spain; 96087@parcdesalutmar.cat

**Keywords:** breast cancer, HER2-positive, targeted therapy, trastuzumab, pertuzumab, resistance, label-free proteomics, bioinformatics

## Abstract

HER2 (human epidermal growth factor receptor 2) is overexpressed in approximately 15–20% of breast cancers, leading to aggressive tumour growth and poor prognosis. Anti-HER2 therapies, such as trastuzumab and pertuzumab, have significantly improved the outcomes for patients with HER2-positive breast cancer by blocking HER2 signalling. However, intrinsic and acquired resistance remains a major clinical challenge, limiting the long-term effectiveness of these therapies. Understanding the mechanisms of resistance is essential for developing strategies to overcome it and improve the therapeutic outcomes. We generated multiple HER2-positive breast cancer cell line models resistant to trastuzumab and pertuzumab combination therapy. Using mass spectrometry-based proteomics, we conducted a comprehensive analysis to identify the mechanisms underlying resistance. Proteomic analysis identified 618 differentially expressed proteins, with a core of 83 overexpressed and 118 downregulated proteins. Through a series of advanced bioinformatics analyses, we identified significant protein alterations and signalling pathways potentially responsible for the development of resistance, revealing key alterations in the protein metabolism, mitochondrial function, and signalling pathways, such as MAPK, TNF, and TGFβ. These findings identify mitochondrial activity and detoxification processes as pivotal mechanisms underlying the resistance to anti-HER2 therapy. Additionally, we identified key proteins, including ANXA1, SLC2A1, and PPIG, which contribute to the tumour progression and resistance phenotype. Our study suggests that targeting these pathways and proteins could form the basis of novel therapeutic strategies to overcome resistance in HER2-positive breast cancer.

## 1. Introduction

Breast cancer is the most common malignancy among women worldwide, accounting for over 2.3 million new cases annually [1]. Despite advances in screening, diagnosis, and treatment, it remains a leading cause of cancer-related deaths in women. One of the most aggressive subtypes of breast cancer is characterised by the overexpression of the human epidermal growth factor receptor 2 (HER2), which occurs in approximately 15–20% of breast cancer cases [2]. HER2-positive breast cancer is associated with rapid tumour growth, higher rates of metastasis, and poorer overall survival compared with those of HER2-negative subtypes [3,4]. The introduction of targeted therapies, particularly monoclonal antibodies, has revolutionised the treatment landscape for HER2-positive breast cancer, leading to significant improvements in patient outcomes. However, the development of resistance to these therapies remains a significant clinical challenge.

The best established and most widely used anti-HER2 therapies involve monoclonal antibodies that target the HER2 receptor. Trastuzumab, a humanised monoclonal antibody, was the first anti-HER2 agent approved for treating HER2-positive breast cancer [5,6]. It binds to the extracellular domain IV of the HER2 receptor, inhibiting receptor dimerisation, and, consequently, the downstream signalling pathways responsible for cell proliferation and survival [7]. Trastuzumab also induces antibody-dependent cellular cytotoxicity (ADCC), recruiting immune cells to attack HER2-overexpressing tumour cells [8]. Pertuzumab, another monoclonal antibody, binds to a different epitope on the HER2 receptor (domain II) and inhibits the dimerisation of HER2 with other HER family receptors, particularly HER3 [9]. The combination of trastuzumab and pertuzumab has become a cornerstone of HER2-positive breast cancer treatment, as the dual inhibition of HER2 leads to a more comprehensive blockade of HER2 signalling pathways, boosting anti-tumour activity [10,11].

Despite the success of trastuzumab and pertuzumab, approximately 15–20% of patients with HER2-positive breast cancer do not respond to these therapies initially, and up to 50% may develop resistance after an initial period of clinical benefit. Resistance to anti-HER2 therapies is now recognised as one of the major barriers to improving outcomes in this population [12]. Resistance to anti-HER2 therapies, whether intrinsic or acquired, is a complex process involving multiple molecular mechanisms [13,14]. These include alterations in the HER2 receptor itself, such as mutations that prevent the binding of monoclonal antibodies or that activate downstream signalling independent of HER2 inhibition. In addition, the activation of compensatory pathways, such as the PI3K/AKT/mTOR pathway, can bypass the blockade of HER2, allowing for continued tumour cell growth despite the presence of HER2 inhibitors. Mutations in the *PIK3CA* gene, loss of the tumour suppressor PTEN, and the overexpression of alternative receptor tyrosine kinases such as HER3 and IGF-1R are among the most commonly involved resistance mechanisms [15].

Moreover, HER2-positive tumours can acquire resistance through alterations in the tumour microenvironment. Factors such as immune evasion, changes in the extracellular matrix, and the presence of immunosuppressive cells can hinder the efficacy of antibody-mediated immune responses [16]. For example, polymorphisms in the Fc receptor genes can impair the ability of immune cells to mediate ADCC in response to trastuzumab treatment. As resistance develops, the effectiveness of anti-HER2 therapies diminishes, leading to disease progression and metastasis. Consequently, understanding the molecular basis of resistance is critical to developing new therapeutic strategies and improving the prognosis of patients with HER2-positive breast cancer.

Proteomic studies have emerged as a powerful tool in the investigation of resistance mechanisms to anti-HER2 therapies [17]. By comprehensively analysing the protein expression profiles of tumour cells, proteomics enables the identification of key proteins and signalling pathways that are altered during the development of resistance. Unlike genomic studies, which focus on DNA instability, proteomics provides direct insights into the functional state of the cell, revealing changes in protein abundance, modifications, and interactions that drive resistance [18]. Proteomic analyses have already uncovered several potential mechanisms of resistance to HER2-targeted therapies. For instance, studies have identified the upregulation of HER3, as well as the activation of downstream effectors in the PI3K/AKT pathway, as common features of resistant tumours [19]. Additionally, proteomic data have revealed the involvement of other receptor tyrosine kinases and signalling pathways, such as MET and AXL, which may be compensatory mechanisms that promote survival in the face of HER2 inhibition [20]. Furthermore, proteomic studies are crucial for identifying potential new therapeutic targets [21]. By comparing the proteomic profiles of resistant and sensitive tumours, researchers can identify proteins that are uniquely altered in resistant cells, providing a basis for developing novel therapeutic interventions. For example, inhibitors targeting the PI3K/AKT/mTOR pathway or HER3-directed therapies are currently being explored in clinical trials of potential strategies to overcome resistance [19].

Advanced bioinformatic tools are crucial for processing and interpreting complex proteomic data, allowing researchers to identify differentially expressed proteins (DEPs), enriched pathways, and protein–protein interactions [22]. Machine learning algorithms and artificial intelligence (AI) are increasingly being applied in bioinformatics to predict resistance mechanisms and identify biomarkers for therapeutic response. For instance, clustering algorithms can group tumours based on their proteomic profiles, revealing distinct subtypes of resistance that may require different therapeutic approaches [23]. In addition to aiding the identification of resistance mechanisms, bioinformatic analyses are critical to the discovery of new therapeutic markers. Proteins identified through proteomic studies can be validated as potential biomarkers for predicting the response to therapy or for monitoring resistance during treatment [24]. This information is invaluable for personalising treatment strategies and improving the clinical management of patients with HER2-positive breast cancer.

In our laboratory, we developed HER2-positive breast cancer cell lines resistant to trastuzumab and pertuzumab to investigate the mechanisms driving resistance. Using label-free mass spectrometry (MS)-based proteomics, we identified 618 DEPs between the sensitive and resistant cell lines. Our bioinformatic analyses revealed significant alterations in the protein metabolism, mitochondrial function, and key signalling pathways, including MAPK, TNF, and TGFβ. Notably, we observed an enrichment in ribosomal proteins and mitochondrial components, indicating a shift in energy production and protein synthesis, which are crucial for cell survival under continuous anti-HER2 therapy. Additionally, resistance was linked to processes such as oxidative stress response and cellular detoxification. These findings indicate that mitochondrial dysfunction, enhanced protein metabolism and detoxification processes are central to resistance, suggesting potential new therapeutic targets to overcome resistance to trastuzumab and pertuzumab.

## 2. Results

In a recent study by our research group, we established four novel HER2-positive BCCLs through extended exposure to trastuzumab and pertuzumab, and assessed their resistance rates [25]. The primary aim of that investigation was to elucidate the underlying mechanisms contributing to resistance to anti-HER2 therapies. We then performed a proteomic analysis of one of the parental vs. resistant BCCL pairs, using label-free MS/MS to identify and quantify the potential biomarkers involved in the development of resistance to trastuzumab and pertuzumab. The SK-BR-3 cell line was selected because it is already well characterised, and so an ideal model for studying HER2-positive breast cancer. SK-BR-3 cells naturally overexpress the HER2 protein, which allows them to respond to HER2-targeted therapies. This enables the study of drug efficacy and the mechanisms of action. Additionally, the cell line has a well-characterised genetic profile, which allows the findings to be correlated with known genetic pathways. Furthermore, it offers a comprehensive set of comparative data and research protocols that facilitate cross-study comparisons and validation.

### 2.1. Adaptive Advantages in SK-BR-3 Breast Cancer Cells: Proteasome, Ribosome Dynamics, and Signalling Pathway Alterations Drive Tumour Progression

An additional objective of this study was to elucidate the distinctive protein expression profile of the SK-BR-3 cell line in the context of the human proteome. A label-free liquid chromatography–MS/MS analysis was conducted. Protein extracts from each cell line were analysed in triplicate. To facilitate a comparison of protein abundance between samples, the data were normalised, enabling the application of an appropriate correction factor for each sample, thereby ensuring that the total protein amount was consistent across all samples (Appendix A). In contrast, the Volcano plot depicts the outcomes of the statistical tests implemented (the log *p*-value versus log_2_ratio or fold-change, FC) of the calculated protein quantity (Appendix A). Proteins situated within the coloured boxes are regarded as DEPs, as they satisfy the conditions of a *p*-value < 0.05 and an FC ≥ 0.6 (1.5 times the original value) for overexpressed proteins or ≤−0.6 for underexpressed proteins. We identified 4239 proteins, 4116 of which could be quantified. Next, we decided to analyse the biological processes, molecular function, and cellular component in which the 4239 identified proteins were considered in a GO over-representation analysis. The use of the PANTHER database classification tool facilitated the organisation of identified proteins according to the phylogenetic trees of gene families. The phylogenetic trees were used to annotate the proteins based on their functional classification, pathway assignment, protein family, orthologous relationships, and paralogous connections. The initial analysis of the identified proteins’ ontological categories revealed an over-representation of proteins primarily associated with ribosomes, mitochondria, and proteasomes (Table 1). In general, the two SK-BR-3 strains, the parental and its resistant derivative, demonstrated an increase in the activity of processes related to protein synthesis (translation, modification, labelling, and complex formation), in cytoplasmic and mitochondrial ribosomes, as well as in their degradation in proteasomes. Additionally, high levels of respiratory and metabolic functions are evident, which are indicative of the cells’ intense proliferative activity. In order to provide a comprehensive visual representation, the GO terms obtained from the 4239 proteins identified in the proteomic analysis were summarised using Revigo. The most prevalent biological processes were transcription, mitochondrial activity, and protein localisation (Figure 1A). Proteasome and RNA binding, and the metabolism were among the most important molecular functions identified (Figure 1B). Finally, as with the over-represented cellular components, we identified mainly the proteins of proteasomes, mitochondria, and ribosomes (Figure 1C).

With regard to the most representative signalling pathways of SK-BR-3 in the context of the total *Homo sapiens* database (Table 2), it was possible to identify four major clusters of proteins that exhibit an increase in both number and abundance (Figure 2A). The first category comprises pathways related to the protein metabolism, and encompasses protein translation and synthesis at the cytoplasmic and mitochondrial levels, as well as degradation by the proteasome. The second category includes pathways associated with oncogenic processes, such as glycolysis, RAS, EGFR, FGF, and CCKR. The third category pertains to DNA synthesis and replication. The final category includes pathways involved in communication and connection between the cytoskeleton and the extracellular matrix. The most relevant classes of proteins in SK-BR-3 were consistent with the following four groups (Figure 2B): (i) proteins involved in the protein metabolism, RNA and ribosome processing and metabolism, translation factors, chaperones, etc.; (ii) proteins involved in the general metabolism, oxidoreductases, and transmembrane signalling receptors; (iii) zinc finger transcription factors, gene-specific transcriptional regulators, and DNA-binding transcription factors; (iv) transmembrane linkers between the extracellular matrix and the actin cytoskeleton (integrins, cadherins, and others).

The over-representation analysis of the 4239 proteins, performed in Reactome, found 3133 identifiers and 2080 pathways that were hit by at least one identifier. Table 3 shows the 25 most relevant pathways ranked by the *p*-value. In conclusion, the results of this analysis corroborate those of the preceding analysis regarding the pivotal role of ribosomal factors, mitochondria, and the functions associated with these organelles. The molecular functions and pathways are primarily connected through processes related to RNA synthesis and processing, as well as by protein translation and production (Figure 3). The Reactome database identified several key relationships, including the RNA metabolism, mRNA and rRNA processing, ribosome formation, translation initiation and regulation, mitochondrial translation, as well as the cellular response to stress and stimuli. In summary, these pathways and functions are inter-related, and are characteristic and essential in tumour cells, conferring adaptive advantages upon them relating to their progression.

The subsequent analysis was conducted using Proteomaps, which illustrates the quantitative composition of proteomes, with a particular emphasis on the changes affecting protein function. Therefore, the proteomap of the representative proteins of SK-BR-3 exhibited an increase in genetic and environmental information processing, biosynthesis, glycolysis, and vesicular transport across diverse organelles (Figure 4). This information enables the rapid and visual capture of cellular physiology, in accordance with the findings of previous analyses.

Ultimately, a gene set analysis (GSEA) of the total protein content in SK-BR-3 enabled the enrichment to be investigated through a comparison of the resistant and sensitive cell populations. The following findings are worthy of note in the context of the resistant lineage: (a) in the context of “all gene sets” (Human MSigDB All gene sets), alterations in the DNA and histone methylation patterns, as well as an increase in the level of regulatory processes mediated by cytokines, particularly vesicular communication, were observed; (b) in the context of “hallmark” gene sets (Human MSigDB Collection H), the interferon response and peroxisome were identified; (c) in the context of “cancer-related” gene sets (Human MSigDB Collections C4, C6), proteins with roles in apoptosis, epithelial-to-mesenchymal transition (EMT), signal transduction in the PI3K-AKT-mTOR pathways, KRAS, increased expression of ErbB receptors and their ligands, and cell cycle control were identified. The protein core that explains this gene set enrichment signal (leading-edge analysis, LEA) demonstrated that the majority of kinases, oncogenes, and a lower, yet still substantial, proportion of cytokines and growth factors were overexpressed in the resistant strain. The most prominent of these were CDK4, CDK6, CDKN2A, NFKB1, ANXA1, SAMHD1, ISG15, STAT2, RAC2, RB1, BCL2, S100A4, and TACSTD2.

### 2.2. Protein Expression in Resistant Cells Revealed Metabolism, Protein Synthesis, and Genetic Regulation Mechanisms of Anti-HER2 Resistance

Next, we investigated the differences in the protein abundance between the SK-BR-3-sensitive cell line and its corresponding SK-BR-3.rTP-resistant counterpart in the absence of anti-HER2 treatment. A total of 618 proteins were differentially expressed with a *p*-value < 0.05, where 349 and 269 of which were, respectively, upregulated and downregulated in the resistant relative to the sensitive cell line. A GO over-representation analysis offered a first glimpse about the biological processes, molecular function, and cellular component in which these differential proteins were involved. The most relevant biological processes included a wide variety of metabolic and ribosome-related processes (Figure 5A). They are all involved in fundamental cellular processes such as ribosome formation, protein transport, metabolism, and the organisation of cellular structures (Table 4). These processes are of critical importance to the functioning and survival of cells in the context of environmental aggression, such as the TP-prolonged treatment. With respect to molecular functions, the list of GO terms emphasises functions involved in DNA binding and transcription regulation, enzymatic activities (especially redox reactions), and signal transduction processes within cells (Figure 5B). A large number of terms focus on proteins that interact with regulatory regions of DNA, specifically in the context of transcriptional regulation and gene expression. In addition, the cell activated biochemical reactions involving oxidation–reduction processes. A third group of proteins transmit signals across cell membranes. These activities are key to cell communication and the signalling pathways that control cellular responses to environmental stimuli. The GO analysis related to cellular components revealed terms involved in structural elements, like the cytoskeleton, organelles (mainly mitochondria), ribosomal subunits, and the nucleolus (Figure 5C). These components play critical roles in maintaining the cell structure, energy production, and protein synthesis. In conclusion, these terms are consistent with the biological processes and functions highlighted in the previous GO analyses, reinforcing the importance of activities associated with ribosomes and mitochondria.

Subsequently, a pathway over-representation analysis was conducted using the Reactome tool. This identified 1377 biological pathways that were significantly enriched in the 618 proteins in the cell-resistant dataset (Figure 6). These pathways encompass key areas of the cellular metabolism, protein synthesis, stress management, and cell death, reflecting a wide array of essential biological processes. These pathways are probably related to the stress experienced by the cells in the process of TP-resistance acquisition. In Table 5, we summarise the most highly regulated pathways. The GO terms encompass critical processes underlying cancer progression, the stress response, and essential cellular metabolism. In particular, these terms reflect molecular mechanisms related to the regulation of the cell cycle, gene regulation, DNA repair, apoptosis, the evasion of senescence, mitochondrial function, cholesterol regulation, and metabolic pathways. It appears that the disruption of cell cycle regulation is a prominent feature of these cells, particularly in the context of RB1 defects that result in uncontrolled cell proliferation and the evasion of oncogene-induced senescence. Similarly, defects in the p16INK4A-CDK4/6 interaction contribute to tumour growth by allowing cells to circumvent normal growth arrest. Membrane receptor signalling serves as a trigger for downstream pathways, including NFKB activation, which plays a significant role in cancer by regulating genes involved in cell survival, proliferation, and metastasis. Furthermore, the regulation of gene expression by hypoxia-inducible factor and methylation provides insight into the mechanisms of cellular adaptation to stress and epigenetic control. The intrinsic pathway of apoptosis is initiated by proteins such as BAK and BAX, which promote alterations in mitochondrial apoptotic factors, facilitate the translocation of death-related proteins to mitochondria, and activate anti-apoptotic BCL-2 members, thereby inhibiting mitochondrial-mediated cell death. Mitochondrial protein import is essential for the maintenance of mitochondrial function, while cholesterol biosynthesis is subject to strict regulation, which ensures optimal cellular cholesterol levels, which are crucial for maintaining membrane integrity and metabolic processes. Finally, metabolic processes such as the peroxisomal lipid metabolism and nicotinamide salvaging ensure an appropriate energy balance and cellular homeostasis.

An in-depth examination of the unique proteome of SK-BR-3.rTP-resistant cells was then conducted, with a particular emphasis on the protein abundance and functional analysis. To achieve this, we employed the Proteomaps tool to visualise the composition of the DEPs between the sensitive and resistant counterparts, focusing on protein abundances and functions. The results are presented in Figure 7. In the initial functional resolution layer, the predominant categories corresponded to metabolic activity (in orange–brown shades) and proteins involved in the stages of the central dogma that transform DNA into proteins (termed “genetic information processing”, in blue). With respect to the metabolism, the highest percentage corresponded to the synthesis, modification, and transport of amino acids, as is expected in cells with a high proliferative rate. This is, in turn, consistent with the functions highlighted in the category of genetic information processing, such as ribosomes, translation factors, protein processing, and chaperones. A third representative category includes signalling, especially the interaction between molecules for the transmission of these signals. This enrichment was particularly notable in various signalling pathways, including the MAPK, TNF, and TGFβ-related pathways.

GSEA was employed to determine and identify sets of genes with statistically significant differences between the SK-BR-3 and SK-BR-3.rTP1 cell lines. A GSEA of 618 DEPs was conducted in various contexts, including all gene sets, hallmarks of cancer, and cancer-related gene sets. In the context of “all gene sets”, the results were deeply intertwined with various biological processes, including gene expression patterns and regulatory mechanisms. Moreover, some gene sets were linked to specific signalling pathways, such as the MAPK8 (JNK) pathway and MYC targets, which are crucial for understanding the signalling mechanisms that regulate cellular functions. In the case of “hallmark” gene sets, we identified proteins involved in the transition of epithelial cells to mesenchymal stem cells, as well as proteins targeted by E2F transcription factors, which are essential for cell-cycle regulation and DNA replication. These proteins play a significant role in cell proliferation and cancer progression. Additionally, proteins that mediate peroxisome formation and function, proteins that mediate programmed cell death, and proteins that encode components of the apical junction complex were identified. Some of these proteins are involved in the process of protein secretion, which is essential for signalling. Furthermore, proteins targeted by the MYC oncogene, which is involved in cell growth, proliferation, and apoptosis, were also identified. In the context of “cancer-related” gene sets, our findings underscored the significance of cell cycle checkpoints and the role of various cyclins and transcription factors in regulating cell division. Furthermore, our findings highlighted the significance of proteins that facilitate tumorigenesis and those engaged in pivotal signal transduction pathways. These gene sets affect a number of cellular processes, including growth, survival, and apoptosis (AKT/mTOR, MEK, and TBK1). Furthermore, they regulate pivotal proteins, such as p53 and TBK1, which maintain cellular homeostasis in the presence of stress. Moreover, they regulate transcriptional regulators of gene expression. The aforementioned factors were found to be related to alterations in proliferation, aberrant differentiation, apoptosis, signal transduction, and transcription repression. Table 6 presents the most pertinent gene sets from the 618 DEPs in the C4 and C6 cancer-associated collections. Finally, a LEA analysis was performed to identify the core set of proteins responsible for the observed enrichment signal within the context of the entire dataset. The most relevant genes in the leading-edge subset of the associated significant gene sets included ANXA1, SLC2A1, S100A4, ISG15, GSTP1, GSN, LGALS1, LCP1, CAPG, HRG, NFKB1, SP1, GLRX, RB1, ECH1, CRABP2, ITGB6, and MVP.

We subsequently analysed the STRING database, which predicts protein–protein interactions, to assess potential associations between the DEPs following the acquisition of resistance. The initial analysis examined the list of 618 upregulated proteins in SK-BR-3.rTP in the context of the 4239 proteins identified in the cell line. In this manner, the use of our own dataset as the statistical background allowed STRING to enhance the precision and relevance of the functional enrichment results while circumventing potential biases introduced by comparisons with the entire protein pool. Our findings revealed an enrichment in many processes, including those related to genomic information processing, mitochondrial functioning, and metabolism, which is consistent with our previous analyses (Figure 8A). Specifically, the SK-BR-3.rTP-resistant cell was found to be enriched in histones, ribosomal proteins (such as RPLs/PRSs, which are indispensable for protein synthesis), and RNA regulators (of splicing, RNA stability, and protein expression modulators). With respect to mitochondrial proteins, we identified components of the mitochondrial ribosomes (MRPSs), which are involved in mitochondrial protein translation, as well as proteins of the internal membrane (TIMMs), which are responsible for importing other proteins from the cytosol. Additionally, we found hubs of cellular energy production, including NDUFA proteins, which are involved in the mitochondrial electron transport chain and ATP generation. These protein families play essential roles in cellular biology, including protein synthesis, gene regulation, and energy production within the cell. A STRING analysis of the most representative proteins of these processes, which are the leading core of the gene sets, revealed the enrichment of functions relevant to tumour progression, including RNA processing, transcription, and translation regulation, as well as the metabolism and cellular response to stimuli (Appendix A). The majority of these proteins are extracellular, many being secreted via exosomes. Of particular note is a small core of microRNA-regulated proteins, which also represent important nodes in altered functions in cancer, including ErbB2, NFkB1, and Notch2 (Figure 8B).

### 2.3. Mitochondrial and Detoxification Pathways Represent Pivotal Mechanisms Underlying Resistance to Anti-HER2 Therapy

When we applied robustness criteria in the analysis of peptide spectrum identifications and quantifications (Mascot score > 70, at least one unique peptide, *p*-value < 0.05, abundance rate > 1.5, and exclusion of unspecific, highly abundant proteins), both lists became more restricted. Specifically, we found 83 more abundant proteins and 118 less abundant proteins in the resistant strain.

The ontology analysis revealed that the overexpressed proteins exhibited enriched mitochondrial components, antioxidant activity functions, and cellular detoxification processes (Figure 9A). Additionally, the proteins demonstrated an involvement in response to inorganic substances, the regulation of metal ion transport, and homeostatic regulation processes. In contrast, the GO analysis of underexpressed proteins yielded less statistically significant results. Nevertheless, the functions related to the regulation of the small-molecule metabolism, catalytic activities, kinases, and protein assembly were identified.

The GSEA and LEA analyses yielded only a limited number of significant results, given the relatively few gene sets analysed. In any case, resistant cells showed significant enrichment in the processes of cytoskeleton reorganisation, reorganisation of cellular complexes of proteins with nucleic acids and lipids, and metabolic processes, which corroborates previous findings. In particular, the most significant genes involved in these processes were ANXA1, GSN, RAC2, LCP1, STOM, and NFKB1 (Figure 9B).

A STRING analysis demonstrated that, with respect to the overexpressed proteins, there were significant enrichments in the proteins associated with responses to toxic substances, ribosome formation, mitochondrial activity, and those associated with the metabolism (oxidative response to stress) (Figure 9C; Table 7). The selection of these overactivated processes makes sense in the context of more proliferative cells, such as SK-BR-3.rTP-resistant cells, being exposed to continuous treatment. These processes produce a change in cell activity, including movement, secretion, enzyme production, gene expression, and so forth, as a consequence of the toxic stimulus of anti-HER2 therapy. Conversely, we identified approximately 100 proteins whose abundance was reduced in the resistant strain, including a considerable number of small proteins associated with the metabolism, particularly oxidoreductase activity, amino acid degradation, and nucleotide metabolism. Additionally, a few other proteins are components of structures, including the peroxisome protein node, ribonucleoproteins of the 7SK snRNP complex, cell–substrate junction proteins, and proteins involved in actin capping. It is important to note that these proteins were scarce in resistant cells, which is consistent with the cellular findings of increased cell proliferation, increased drug resistance, and increased invasiveness.

The goal of this study was to elucidate the core mechanisms underlying anti-HER2 resistance. To this end, we used Metascape to analyse the most significant proteomic data comparing therapy-sensitive and therapy-resistant SK-BR-3 cell lines. Metascape’s comprehensive suite of tools enabled us to functionally characterise 83/118 of the protein datasets at different levels. These included pathway enrichment analyses, which identified the key biological pathways implicated in resistance, and ontology analyses, which revealed over-represented biological processes, cellular components, and molecular functions. This provided us with a broad understanding of the cellular mechanisms involved. Protein–protein interaction networks were employed to identify the central proteins and regulatory networks that drive resistance. Cluster analysis was used to highlight potential regulatory mechanisms performed by groups of co-regulated proteins. Functional enrichment analysis in the resistant cell line uncovered specific functions associated with resistance. Finally, a comparative analysis with existing datasets validated our findings and provided additional context. A Metascape analysis of the 83 overexpressed proteins revealed the enrichment of a wide variety of categories, including ontology terms, signalling pathways, hallmark gene sets, structural complexes, signature modules, and transcription factor targets. The most prominent processes were those related to ribosome synthesis, the response to oxidative stress, the control of homeostasis, ion transport, and the regulation of the protein metabolism at the cytosolic and mitochondrial levels (Figure 10A). These results are consistent with and complement the previous GO analysis. Furthermore, densely connected network components were identified through the study of protein–protein interactions (Figure 10B). These groups of proteins served as reference nodes with which to identify the pathways and processes with functional relevance and biological significance within the network, including ribosome formation, homeostasis control, and neutrophil trap formation. With regard to the proteins that were reduced in the resistant line, we observed intriguing alterations in processes that, in some way, served as compensatory mechanisms to those corresponding to the most strongly overexpressed proteins (Figure 10C). The enrichment revealed the involvement of protein catabolic processes, along with alterations in the metabolism of other molecules (cofactors, vitamins, and organic acids), integrin-mediated cell adhesion, and modifications in three major signalling pathways (VEGF, MAPK, and FAS). It should be noted that enrichment in these downregulated functions would be considered a loss of activity in most cases. All of the protein–protein interactions among the input proteins were extracted from the PPI data source and used to form a network (Figure 10D), which revealed three protein subsets. One of these, comprising almost 20 network proteins, was associated with processes that regulate protein localisation, as well as alterations of signal transduction by growth factor receptors and second messengers. The other two hubs were related to the metabolic processes of proteins and other molecular types.

### 2.4. The Clinical Significance of Protein Sets in Terms of Survival Analysis and Biomarker Validation: Implications for Prognosis and Treatment

The modifications to the processes and proteins identified in our analyses may prove significant in the long-term evolution of the disease. The cellular management of drug stress alters the regulation of metabolism and homeostasis, which, in turn, gives rise to changes in the principal intracellular signalling pathways. Modifications to these pathways may activate cell survival and proliferation mechanisms, thereby counteracting the effects of treatments and ultimately producing treatment resistance. Modifications to the proteins involved in therapeutic resistance can significantly affect breast cancer patient survival. These modifications have implications not only for the immediate response to treatment, but also for the long-term management and prognosis of breast cancer. To assess the correlation between the gene expression levels of the identified proteins and clinical outcomes, we employed the online survival analysis tool Kaplan–Meier Plotter [26], which encompasses a vast repository of gene expression data and survival information derived from over 15,000 patients across various databases, including The Cancer Genome Atlas (TCGA), Gene Expression Omnibus (GEO), and the European Genome-phenome Archive (EGA). A meta-analysis of the data obtained from the MS/MS label-free study of the resistant line SK-BR-3.rTP was carried out to determine the potential validity of the selected candidates as prognostic markers. The impact of the most relevant proteins on survival was analysed. Some of them were found to significantly discriminate between patient outcomes based on expression levels (Figure 11). In particular, the analysis of the SLC2A1 and PPIG proteins (Figure 11A,B) demonstrated that the difference in RFS between the high- and low-expression groups was statistically highly significant, whereby patients with a low level of expression were at approximately half the risk of recurrence compared with high-expression patients. Therefore, SLC2A1 and PPIG both appear to be significant prognostic markers in patients with HER2-positive breast cancer.

Furthermore, in order to assess the prognostic capacity of a molecular signature, we attempted to identify the proteins that were filtered in our analyses and the determinants of the resistant phenotype. The set of six differentially overexpressed proteins (CCDC43, EIF4EBP2, MRPS15, NDUFV3, PPIG, and YIPF5) had a high prognostic value for RFS (HR = 0.36, *p*-value = 8 × 10^−12^; Figure 11C) and OS (HR = 0.56, *p*-value = 0025; Figure 11D). In the context of our cellular resistance model, these data suggest that intracellular adaptation to therapeutic stress and molecular and metabolic reconditioning may be factors contributing to the acquisition of cellular resistance. By refining the analytical methodology, prognostically useful information can be extracted from the PEDs, thereby enabling the discrimination of patients based on their molecular signatures. It can therefore be surmised that implementing specific treatments directed against these targets and pathways could notably improve the survival prognosis.

### 2.5. Identification of Therapeutic Candidates for Reversing Resistant Oncoproteome Signatures in SK-BR-3.rTP Cells Using Computational Screening of Small-Molecule Perturbagens

In order to identify the small-molecule drugs that align with the selected oncoproteome, we conducted an iLINCS connectivity map analysis to identify the most precise therapeutic candidates. This analysis aimed to identify the compounds capable of reversing the resistant oncoproteome signature of the SK-BR-3.rTP cell line (negative Z-score). Consequently, 408 LINCS chemical perturbagen signatures (234 positively correlated and 174 negatively correlated) were matched to the query signature with an FDR < 0.01 (Appendix A). The 20 most strongly negatively correlated compounds, ranked by connectivity Z-scores, are presented in Figure 12. Among these, the CMap with negative correlation predicted specific perturbagens targeting MEK1 (MAP2K1) and MEK2 (MAP2K2), CDKs, EGFR, HSP90, or IRAK1, which could potentially reverse the oncoproteome activity pattern of SK-BR-3.rTP. It is of particular note that PD-0325901 and selumetinib, MEK1/2 inhibitors, were ranked first and third, with FDRs of 1.1 × 10^−166^ and 7.2 × 10^−67^, and Z-scores of 27.86 and 17.76, respectively.

## 3. Discussion

### 3.1. Challenges and Models of Resistance in HER2-Positive Breast Cancer

Resistance to cancer therapy presents a significant clinical challenge, with consequences that include a threat to patient survival and an increase in treatment costs. The development of laboratory models of acquired resistance is imperative if resistance mechanisms and, thereby, novel therapeutic interventions are to be identified. This study generated HER2-positive breast cancer cell lines resistant to trastuzumab and pertuzumab through prolonged exposure. The SK-BR-3 cell line [27], which possesses inherent HER2 overexpression, has been identified as a valuable model for research [28,29]. Its extensive use has produced comprehensive data, facilitating comparative studies and validation. The clinical relevance and responsiveness of SK-BR-3 cells to HER2-targeted therapies renders them essential for evaluating drug efficacy and developing new strategies.

### 3.2. Proteomic Insights into Molecular Challenges in the SK-BR-3 Cell Line

Proteomic approaches facilitate the comprehensive analysis of protein-level changes in cell lines and their resistant variants, thereby revealing drug resistance mechanisms, potential therapeutic targets, and biomarkers for predicting treatment responses [30]. These insights serve to complement genomic studies, revealing the functional consequences of resistance and providing a foundation for the development of personalised therapies that aim to overcome or prevent resistance in HER2-positive breast cancer [31,32]. The proteomic and bioinformatic analyses of SK-BR-3 and its resistant variant identified significant alterations in proteins, cellular processes, and signalling pathways directly associated with the oncogenic characteristics of the cell lines. These changes highlight critical pathways involved in tumorigenesis, including dysregulated protein synthesis, metabolic reprogramming, and enhanced survival signalling, all of which contribute to the aggressive, cancerous phenotype observed in parental and drug-resistant SK-BR-3 cells. From the 4239 proteins identified overall, key findings revealed an upregulation of proteins involved in ribosomal function, mitochondrial activity, and proteasome dynamics, suggesting increased protein synthesis, degradation, and metabolic activity in resistant cells. The GO analysis highlighted over-represented biological processes such as transcription, mitochondrial function, and protein localisation. Pathway analysis identified clusters related to the protein metabolism (translation and degradation), oncogenic signalling (e.g., glycolysis and RAS), and cytoskeletal communication. Further analyses, including those with Reactome and Proteomaps, confirmed the involvement of mitochondrial translation, RNA processing, and stress response pathways, all of which are crucial for tumour progression. Gene set enrichment analyses identified increased activity in the PI3K-AKT-mTOR, apoptosis, and EMT pathways in the resistant cells. Key proteins associated with resistance included CDK4, CDK6, BCL2, and STAT2.

The alterations identified in ribosomal, mitochondrial, and proteasome functions in SK-BR-3 cells are consistent with the findings from other studies that highlight the role of increased protein synthesis and degradation in cancer progression [33]. Protein homeostasis, including heightened proteasome activity, is known to contribute to tumour survival by facilitating the rapid turnover of damaged proteins and supporting oncogenic signalling pathways [34]. The proteomic analysis of SK-BR-3 further highlights the role of oncogenic pathways such as PI3K-AKT-mTOR, which is frequently activated in HER2-positive cancers and linked to treatment resistance [35]. EMT, on the other hand, is a key process in cancer metastasis and drug resistance that enables tumour cells to adopt more aggressive, invasive phenotypes [36]. In conclusion, this first part of our study provided a comprehensive view of how proteomic and metabolic alterations contribute to pathogenesis in HER2-positive breast cancer and highlighted the role of oncogenic pathways such as PI3K-AKT-mTOR, which is frequently activated in this subtype.

### 3.3. Mechanisms of Resistance to Dual HER2 Blockade in the SK-BR-3.rTP-Resistant Cell Line

The identification of 618 DEPs between trastuzumab/pertuzumab-sensitive and resistant SK-BR-3 cell lines (349 and 269 were expressed at higher and lower levels, respectively) provided significant insights into molecular alterations associated with resistance. These proteomic changes highlight known resistance mechanisms and suggest potential biomarkers or therapeutic strategies. Enrichment analysis revealed that proteins involved in ribosomal function, protein synthesis, apoptosis, and mitochondrial activity were upregulated in resistant cells, indicating that heightened cellular metabolism and protein production act as survival mechanisms under anti-HER2 therapy. This finding is consistent with studies demonstrating that resistance to HER2-targeted therapies frequently involves metabolic reprogramming and enhanced protein turnover [37,38,39]. Resistant cancer cells frequently undergo metabolic reprogramming, increasing the reliance on the glucose metabolism (aerobic glycolysis), fatty acid synthesis, and glutaminolysis to adapt to therapeutic pressure [40]. This metabolic shift fosters accelerated proliferation and survival, despite the intervention of therapeutic agents [34]. The increase in proteins associated with energy production, amino acid synthesis, and mitochondrial function in resistant cells reflects a survival strategy by which increased glycolysis (the Warburg effect) and enhanced mitochondrial function meet the high energy demands of aggressive, resistant tumours [41]. In the context of HER2-positive breast cancer, this metabolic reprogramming enables resistant cells to circumvent the energy stress induced by anti-HER2 therapies, thereby promoting survival and growth.

The phenomenon of resistance to HER2 therapies is associated with elevated protein turnover, a process that facilitates the management of the metabolic stress induced by these treatments. This process encompasses both the enhanced degradation of proteins, predominantly via the ubiquitin–proteasome pathway, and the accelerated synthesis of pivotal proteins that facilitate oncogenic signalling. When subjected to therapeutic stress, HER2-positive cells may upregulate the pathways related to protein homeostasis, which ensures the rapid breakdown of damaged or misfolded proteins while promoting the synthesis of essential growth and survival proteins [42]. These metabolic shifts are pivotal in the development of drug resistance because they enable cancer cells to maintain oncogenic signalling pathways and resist apoptosis, thereby rendering traditional HER2 inhibitors ineffective. A notable finding in resistant cell lines was the increased expression of ribosomal proteins and protein synthesis pathways. This increased protein production is vital for the rapid proliferation and survival of cancer cells under targeted therapy. For instance, resistance to trastuzumab has been linked to dysregulation in translational control mechanisms. Increased protein synthesis in these resistant cells supports their survival by producing oncogenic proteins that maintain HER2 signalling.

The upregulation of pathways such as MAPK, TNF, and TGFβ indicates that oncogenic signalling networks are activated in resistant cells. These pathways, which drive cancer progression, are often hyperactivated in resistant HER2-positive breast cancer. The increased abundance of proteins involved in oxidative stress response and cellular detoxification aligns with known resistance mechanisms that allow cancer cells to enhance their stress response pathways to adapt to therapeutic pressure. Consequently, targeting these pathways could offer novel therapeutic strategies to overcome resistance. Furthermore, the analysis revealed enrichment in biological processes such as the EMT, apoptosis regulation, MYC targets, cell cycle control, and oncogenic signal transduction via PI3K/AKT/mTOR signalling in resistant cells. These processes contribute to therapeutic resistance, because the EMT is associated with increased invasiveness and metastatic potential [43,44], while PI3K/AKT/mTOR pathway activation circumvents the inhibitory effects of HER2-targeted therapies [45,46]. These findings are consistent with those of previous studies that have implicated the EMT and aberrant PI3K signalling in the development of resistance [47].

Our study found the enrichment of proteins involved in DNA binding, transcriptional regulation, and chromatin modification, particularly those regulating the cell cycle and apoptosis pathways. This finding underscores the ability of resistant cells to circumvent conventional growth control mechanisms. Alterations in cell cycle regulation were prominent, with defects in the RB1 and CDK4/6 pathways allowing cells to evade growth arrest and senescence. Rb and E2F1 have been identified as pivotal cell cycle regulators; Rb binds to E2F1 when not hyperphosphorylated, thus preventing transcription. Cyclins activate CDKs, leading to Rb hyperphosphorylation and the release of E2F1 to promote transcription [48,49]. Genomic aberrations in the CDK-Rb-E2F pathway are prevalent in breast cancer, resulting in enhanced resistance to chemotherapy and targeted therapies, particularly in HER2-positive cancers [50]. The dysregulation of cell cycle checkpoints, such as defects in the RB1 pathway, allows resistant cells to proliferate [51]. This finding is consistent with broader-ranging research demonstrating that HER2-positive cancer cells frequently manipulate transcriptional regulators (e.g., E2F and MYC) to drive resistance [52]. Consequently, there have been concerted efforts to pharmacologically block the CDK-Rb-E2F pathway. Selective CDK4/6 inhibitors, including palbociclib, abemaciclib, and ribociclib, have demonstrated efficacy in treating primary HER2-positive tumours and in predicting resistance to HER2-targeted therapies [53]. The present study suggests that this pathway and associated proteins are involved in HER2-positive breast cancer models that do not express hormone receptors. Further investigation and functional validation are required to determine the potential contribution of these proteins to resistance to dual HER2 blockade with trastuzumab and pertuzumab, as well as the use of CDK4/6 inhibitors.

Our study revealed that the cells exhibit resistance to apoptosis, accompanied by the differential regulation of proteins involved in the intrinsic apoptosis pathway, including BCL-2 family members. This finding is of crucial significance, given that HER2-targeted therapies frequently induce cell death in tumour cells. The ability of resistant cells to block apoptosis, often by overexpressing anti-apoptotic proteins, has been identified as a resistance mechanism that helps them survive in the presence of HER2 inhibitors [54].

The analysis of protein–protein interactions among the 618 DEPs revealed that resistant cells exhibited the enrichment of proteins involved in mitochondrial energy production, transcription regulation, and RNA processing. This finding serves to emphasise the pivotal role that the energy metabolism and protein synthesis play in the context of drug resistance. The study highlighted an increase in mitochondrial proteins, particularly those involved in the electron transport chain and ATP production, suggesting that resistant cells rely on enhanced oxidative phosphorylation to meet their energy demands. Key proteins identified, including ANXA1, SLC2A1, S100A4, and NFKB1, contribute to tumour progression and the resistance phenotype. The significance of these findings lies in their being consistent with findings from previous research on adaptive drug resistance mechanisms in HER2-positive breast cancer, emphasising the importance of metabolic reprogramming and protein turnover in developing resistance to HER2-targeted therapies [55,56]. Consequently, our findings indicate the potential for therapeutic interventions targeting metabolic pathways, protein synthesis, and apoptotic regulation.

### 3.4. Core Protein Analysis Revealed Mechanisms of Resistance

A detailed investigation was conducted into the key biological processes altered in trastuzumab/pertuzumab-resistant SK-BR-3 cells in order to understand the mechanisms underlying the resistance to anti-HER2 therapies in breast cancer. It focused on 83 overexpressed and 118 downregulated proteins in the resistant cell line, highlighting cytoskeletal reorganisation, detoxification pathways, and oxidative stress responses as core features of resistance. Proteins such as ANXA1, GSN, and RAC2, which are involved in cytoskeletal changes, enhance cell motility, invasiveness, and adaptability to the toxic environment of continuous anti-HER2 treatment [57]. This finding is consistent with the established role of cytoskeletal dynamics in cancer metastasis and drug resistance, where they facilitate tumour cell survival by evading cell death and resisting therapy-induced apoptosis [58]. Furthermore, the results emphasised the significance of mitochondrial function, particularly in the context of detoxification and oxidative stress responses. The upregulation of proteins associated with antioxidant activity and mitochondrial detoxification reflects an adaptive mechanism in which resistant cells mitigate damage from reactive oxygen species (ROS) during anti-HER2 therapy [59]. This finding is corroborated by the existing body of literature, which demonstrates that cancer cells experiencing oxidative stress upregulate mitochondrial ROS scavenging systems to evade apoptosis, thereby contributing to therapy resistance. It is therefore vital to emphasise that detoxification pathways and oxidative stress regulation are crucial for maintaining cellular homeostasis under continuous HER2-targeted treatment.

Conversely, the 118 downregulated proteins indicate a compensatory reduction in metabolic activities, such as amino acid degradation and nucleotide metabolism, which are necessary for proliferative cells [60,61]. Downregulated signalling pathways, including VEGF and MAPK, suggest a reduction in pro-survival signalling, allowing resistant cells to redirect resources towards coping with toxic stress [62]. These shifts in cellular activities are specific to the core set of proteins, and differ from the broader metabolic and proliferative processes observed in the 618 proteins. This focus on stress management, detoxification, and cytoskeletal changes, rather than generalised proliferation or metabolism, reflects advances in our understanding of cancer therapy resistance. Studies increasingly show that resistant cells undergo profound metabolic rewiring and adapt structurally to persist under drug pressure [63]. Our new findings indicate that targeting mitochondrial detoxification and cytoskeletal dynamics could be the basis for novel therapeutic strategies for overcoming HER2 resistance [64].

### 3.5. Leveraging Molecular Markers to Overcome Therapeutic Resistance in Breast Cancer

The use of molecular markers of resistance facilitates the real-time monitoring of treatment response, enabling dynamic therapy adjustments and boosting success rates. These markers can identify patients at high risk of developing resistance, thus enabling rigorous surveillance and the early modification of personalised therapeutic strategies. This approach may result in the development of new targeted therapies that circumvent resistance, thereby increasing long-term survival rates. Patients exhibiting high levels of expression of SLC2A1, PPIG, or proteins in the prognostic molecular signature have been shown to have reduced recurrence-free survival, which indicates that they are suitable for therapeutic intervention or patient stratification.

A comprehensive understanding of the molecular underpinnings of resistance can help us develop more precise and effective therapeutic interventions. Identifying key proteins, such as those delineated in the present study, facilitates the design of targeted therapies that can overcome resistance and enhance survival rates. Therapeutic resistance in breast cancer has serious implications, including rapid cancer progression, more aggressive stages, and a decline in quality of life. Even after initial treatment success, the presence of resistant cancer cells can lead to recurrence, which is more difficult to treat and negatively affects long-term survival. Furthermore, the development of resistance can favour the dissemination of cancer to other organs, thereby complicating treatment regimens and reducing the likelihood of a complete cure.

### 3.6. Identification of Drug Candidates to Overcome HER2-Positive Breast Cancer Resistance

The iLINCS analysis for CMap was carried out to compare the transcriptional signatures of the resistant state with those of chemical perturbations, with the objective of identifying potential drug candidates [65]. This analysis, performed on the 618 DEPs, aimed to identify drugs capable of reversing resistance-associated changes by inhibiting or blocking activated signalling pathways. Using the extensive LINCS dataset, which includes transcriptional responses to various chemical and genetic perturbations, the analysis identified several promising molecules.

The most significant molecule identified was PD-0325901, which targets MAP2K1 and MAP2K2. A similar approach was adopted in the review of CI-1040, an analogue targeting the MAPK pathway in HER2-positive breast cancer [66]. The preclinical studies conducted so far have indicated that both drugs have the potential to impede downstream signalling, although concerns about toxicity have also been raised. Selumetinib, another potential candidate targeting MAP2K1 and MAP2K2, has demonstrated efficacy in suppressing cell proliferation and migration in various cancers, including HER2-positive breast cancer [67]. Its potential to enhance existing treatments by virtue of its use in combination therapies is also highlighted.

The analysis also identified CDK inhibitors as significant disruptors. Dinaciclib, a drug that targets CDK1, CDK2, CDK5, and CDK9, has demonstrated efficacy in the treatment of breast cancer by reducing the expression of stem cell markers and decreasing cancer cell viability [68]. Dinaciclib is currently undergoing clinical trials, and initial findings have demonstrated its safety and anti-tumour activity. Dinaciclib is one of several CDK inhibitors that have been developed to target multiple CDKs, and that offers a comprehensive strategy to disrupt cell cycle progression and potentially overcome resistance [69].

In preclinical models of HER2-positive and trastuzumab-resistant breast cancer, HSP90 inhibitors, including NVP-AUY922 and a carbamate derivative, have demonstrated significant efficacy by inhibiting HSP90AA1 [70]. This results in the destabilisation of multiple oncogenic proteins, leading to cancer cell death. The combination of HSP90 inhibitors with other treatments has the potential to enhance the overall efficacy by targeting multiple pathways involved in cancer progression and resistance.

In conclusion, the in silico approach identified MAP2K1 and MAP2K2 as the most promising targets, and the iLINCS CMap ranked compounds targeting CDKs, HSP90, and TOP2A as the most effective at reversing resistance in the SK-BR-3.rTP cell line. This experimental design could serve as a foundation for developing personalised precision medicine and molecularly guided therapy.

### 3.7. Summary and Implications

Therapeutic resistance in HER2-positive breast cancer poses a significant challenge to the efficacy of standard treatments, including monoclonal antibody therapy. Alterations in proteins regulating the metabolism, stress response, and signalling pathways can accelerate disease progression and reduce treatment response rates. Cancer cells adapt to therapy-induced stress by activating survival mechanisms that manage oxidative stress and repair damage, allowing them to survive and proliferate. Furthermore, changes in signalling pathways and metabolism can facilitate invasion and metastasis, thus complicating treatment and reducing long-term survival chances. The present study demonstrates that resistance to HER2-targeted therapies in SK-BR-3 cells is driven by enhanced mitochondrial activity, detoxification, altered metabolism, and cytoskeletal reorganisation. These adaptations enable cancer cells to evade apoptosis, maintain high proliferation rates, and resist continuous anti-HER2 treatment.

This study primarily utilised in vitro models based on HER2-positive breast cancer cell lines, which do not fully recapitulate the complexity of the tumour microenvironment found in vivo. Factors such as immune cell interactions, extracellular matrix components, and the influence of systemic factors are not adequately represented in these in vitro systems. Consequently, the findings from this study may not entirely reflect the behaviour of tumours in patients. To confirm the relevance and applicability of these results to human breast cancer therapy, further validation using in vivo models and clinical samples is necessary. A more comprehensive understanding of these processes may aid the identification of potential therapeutic targets to overcome resistance in HER2-positive breast cancer. The study of resistance markers is crucial to the development of more effective, personalised therapeutic strategies and the enhancement of prognostic value.

## 4. Materials and Methods

### 4.1. Cell Cultures and Reagents

The human breast cancer cell line SK-BR-3 (HTB-30) was purchased from the American Type Culture Collection (ATCC, Manassas, VA, USA), maintained in cell culture, treated with continuous exposure to trastuzumab and pertuzumab, and assayed, as previously described [25,71].

### 4.2. Proteomic Analysis

The MS analysis was conducted as indicated in our previous publication [25]. Briefly, SK-BR-3 and SK-BR-3.rTP cells were cultured, lysed in RIPA buffer plus peptidase and phosphatase inhibitors, and the protein concentration was measured using the BCA assay. Proteins were precipitated, digested with trypsin, and analysed using liquid nanochromatography coupled to a Q-Exactive HF mass spectrometer (Thermo Fisher Scientific, Waltham, MA, USA). MS/MS data were processed with Proteome Discoverer v2.5 (Thermo Fisher Scientific) and MASCOT v.2.8 (Matrix Science Ltd., London, UK), using the UniProt database (UniProt consortium, EMBL-EBI, Hinxton, UK; SIB, Lausanne, Switzerland; PIR, Reston, VA, USA) with taxonomic restriction to humans (release 2023_03). Differential protein expression was assessed with ANOVA, and significant proteins were identified based on specific criteria. Data were deposited in the ProteomeXchange Consortium (PXD045804).

### 4.3. Gene Ontology (GO) Analysis

We performed the functional profiling of the proteomic data using the GO resource from the GO Consortium server (National Human Genome Research Institute, US National Institutes of Health) [72]. Functional enrichment analysis of over-represented ontology terms was performed with the GO Enrichment Analysis tool powered by PANTHER [73]. This allowed us to categorise the molecular function, biological process, and cellular localisation of the unique proteins identified in this study. Furthermore, preliminary data on signalling pathways and protein classes were obtained. Only those terms exhibiting a false discovery rate (FDR) value < 0.05 were deemed statistically significant. Ultimately, extensive lists of GO terms were condensed through the removal of redundant terms with Revigo (Ruđer Bošković Institute, Zagreb, Croatia) [74].

### 4.4. Proteomaps

Proteomaps (INRAE, Paris, France) [75] are a visual representation of the quantitative composition of proteomes, with a particular focus on protein function. They are automatically constructed from proteome data and are based on the KEGG Pathways gene classification. Each protein is shown by a polygon, with functionally related proteins arranged in common regions. To highlight highly expressed proteins, the polygonal areas represent protein abundance, weighted by protein size. The maps visualise three levels of functional categories and two level of individual proteins. Proteomaps were constructed from the dataset using the online tool [76].

### 4.5. Gene Set Enrichment Analysis

Gene Set Enrichment Analysis (GSEA) was performed [77]. Functional enrichment was applied using annotations from the MSigDB, Reactome, KEGG, and NCI databases. In particular, we utilised a range of human cancer-related collections, including ’hallmark’ gene sets (Human MSigDB Collection H), ‘cancer-oriented’ gene sets (Human MSigDB Collection C4), and ‘oncogenic signature’ gene sets (Human MSigDB Collection C6). Genes were ranked based on the limma moderated t-statistic [78]. After Kolmogorov–Smirnoff testing, those gene sets showing an FDR < 0.05 were considered enriched between the classes under comparison.

### 4.6. Protein Interaction Analysis

Protein–protein interactions were analysed using the STRING database (SIB, Switzerland; CPR, København, Denmark; EMBL, Heidelberg, Germany) (version 12.0) [79]. Identified proteins were input into STRING to construct interaction networks, with a confidence score threshold of 0.7 to ensure high-confidence interactions. The analysis included both direct (physical) and indirect (functional) associations, integrating data from various sources, such as experimental data, computational prediction methods, and public text collections. The resulting interaction networks were visualised and further analysed to identify key hubs and interaction clusters, providing insights into the functional relationships and pathways involved in breast cancer cell lines.

### 4.7. Metascape Analysis

Functional enrichment and pathway analysis were conducted using Metascape (Metascape Team, Beijing, China) [80]. Identified proteins were uploaded to Metascape, where they were annotated and analysed for enriched biological processes, molecular functions, and cellular components. The analysis integrated multiple databases, including GO, KEGG, and Reactome, to provide a comprehensive overview of the biological significance of the protein dataset. The enrichment results were visualised through bar graphs and network diagrams, highlighting key pathways and functional clusters relevant to breast cancer biology. Terms with a *p*-value smaller than 0.01, a minimum count of 3, and an enrichment factor greater than 1.5 (the enrichment factor is the ratio of the observed counts to the counts expected by chance) were collected and grouped into groups based on their similarities of membership (subtrees with a similarity greater than 0.3 were considered a group) and corrected using the Benjamini–Hochberg algorithm.

### 4.8. Validation of Survival Biomarkers

We estimated the prognostic value of different protein clusters in HER2-positive breast cancer using an online database, Kaplan–Meier Plotter (www.kmplot.com, accessed on 25 November 2024) (A5 Genetics Ltd., Und, Hungary) [26], which contained the gene expression data and survival information of more than 35,000 cancer patients. Gene expression data and relapse-free (RFS) and overall survival (OS) information are publicly available from the Gene Expression Omnibus (GEO, https://www.ncbi.nlm.nih.gov/geo/), the European Genome-phenome Archive (EGA), and The Cancer Genome Atlas (TCGA). To calculate the survival estimate of a group of candidate markers, patient samples were split into two groups by median expression (high versus low expression) and assessed by a Kaplan–Meier survival plot, with the hazard ratio (HR) with 95% confidence intervals and the log-rank *p*-value.

### 4.9. In Silico Connectivity Map Analysis

In order to reverse the resistant oncoproteome of the SK-BR-3.rTP cell line into the SK-BR-3 state, a set of chemical perturbagens was identified using the pharmacogenomic Connectivity Map (CMap) via the iLINCS database (University of Cincinnati, Cincinnati, OH, USA). The Integrative Library of Integrated Network-Based Cellular Signatures (ILINC) web server (www.ilincs.org, accessed on 21 November 2024) was utilised for this purpose [81]. A query signature of the up- and downregulated genes was constructed based on the data extracted from the DEP analysis, which was conducted to compare the SK-BR-3.RTP and SK-BR-3 cell lines (Appendix A). The query signature was submitted to iLINCS for perturbation connectivity analysis against LINCS chemical perturbagen signatures (143,374 signatures; accessed on 19 November 2024). Chemical perturbagens were considered valid when the FDR < 0.01 and the *p*-value < 0.01.

## 5. Conclusions

The landscape of HER2-positive breast cancer treatment has evolved notably since the advent of anti-HER2 therapies, such as those based on trastuzumab and pertuzumab. Nevertheless, the emergence of resistance represents a significant challenge to long-term clinical success. Proteomic studies have yielded valuable insights into the mechanisms of resistance and have identified potential novel therapeutic targets. The incorporation of bioinformatic analytical techniques has further augmented our capacity to interpret proteomic data and devise strategies to overcome resistance. Further research in this field has the potential to enhance therapeutic outcomes for patients with HER2-positive breast cancer and mitigate the effect of resistance on survival.

In light of the findings of our study, we can conclude that the resistance to anti-HER2 therapies in HER2-positive breast cancer is closely associated with alterations in metabolic reprogramming, protein turnover, and the CDK-Rb-E2F pathway. The proteomic analysis of the SK-BR-3 cell line and its resistant counterpart revealed significant alterations in protein expression associated with gene regulation, ribosome function, mitochondrial activity, and proteasome dynamics. These alterations provide evidence of the adaptive advantages that enable tumour progression and resistance. The enrichment of DEPs (such as ANXA1, SLC2A1, S100A4, NFKB1, GSN, and RAC2) in these pathways serves to illustrate the intricate interplay between factors determining cellular sensitivity and resistance. The study highlights the potential of targeting metabolic pathways, protein synthesis, and apoptotic regulation as therapeutic strategies to overcome resistance. Further functional validation is needed to confirm the roles of these proteins in resistance mechanisms, particularly in the context of dual HER2 blockade with trastuzumab and pertuzumab. Our study exemplifies the need to use proteomics and bioinformatics to elucidate intricate resistance mechanisms, and thereby inform decisions about the direction of future therapeutic development.

This research contributes to the growing body of knowledge about resistance mechanisms in HER2-targeted therapy, emphasising the value of proteomics in identifying potential therapeutic targets for more efficacious future treatments for HER2-positive breast cancer.

## Figures and Tables

**Figure 1 ijms-26-01559-f001:**
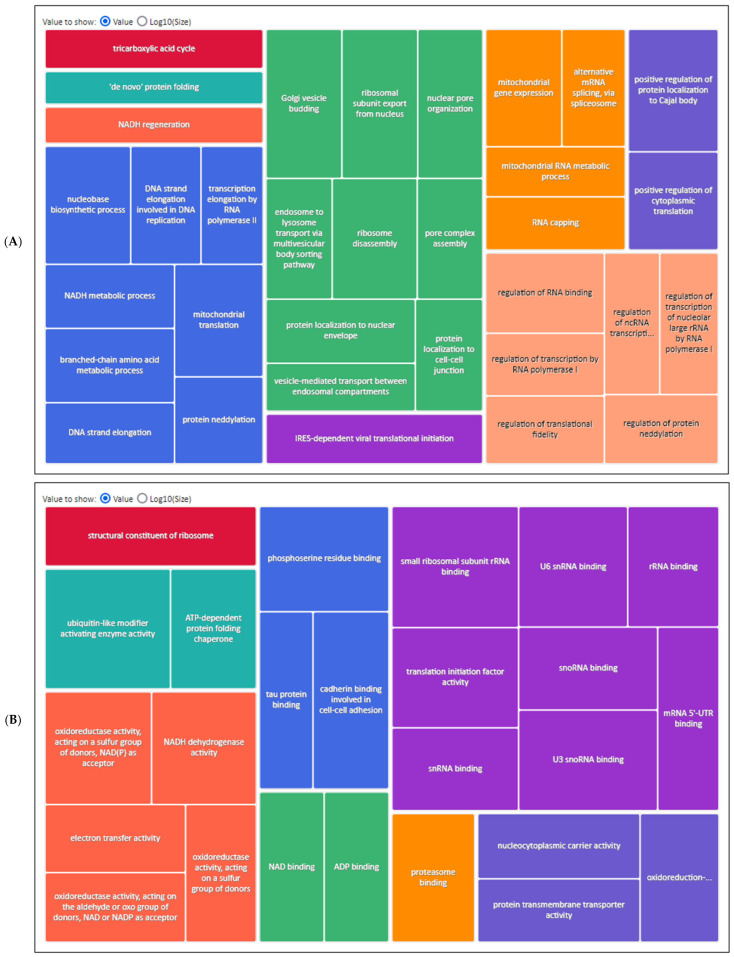
The comprehensive lists of protein annotations derived from the PANTHER ontology analysis, which were subsequently condensed into treemaps through the utilisation of the Revigo algorithm for: (**A**) Biological process; (**B**) Molecular function; (**C**) Cellular component. Charts were constructed using the percentages of proteins annotated with the non-redundant GO terms from Table 1 (FC ≥ 3.0). Each rectangle represents a significant GO term, denoting a supercluster of loosely related terms. The size of the rectangles is proportional to the *p*-value provided by the Blast2GO analysis, whereby a larger rectangle indicates a more significant GO term.

**Figure 2 ijms-26-01559-f002:**
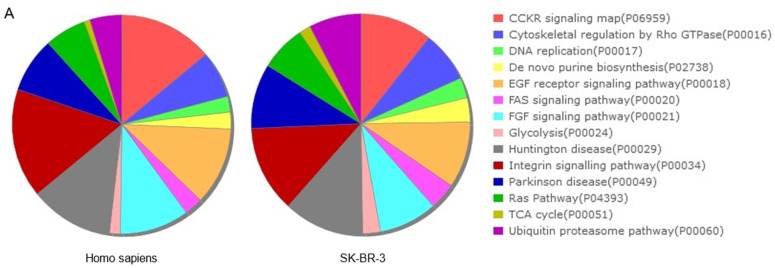
GO-based functional annotation of 4239 proteins of SK-BR-3 as analysed by PANTHER for the following: (**A**) Signalling pathways; (**B**) Class of proteins. FDR < 0.05; FC > 1.5; ratio found/list > 0.33.

**Figure 3 ijms-26-01559-f003:**
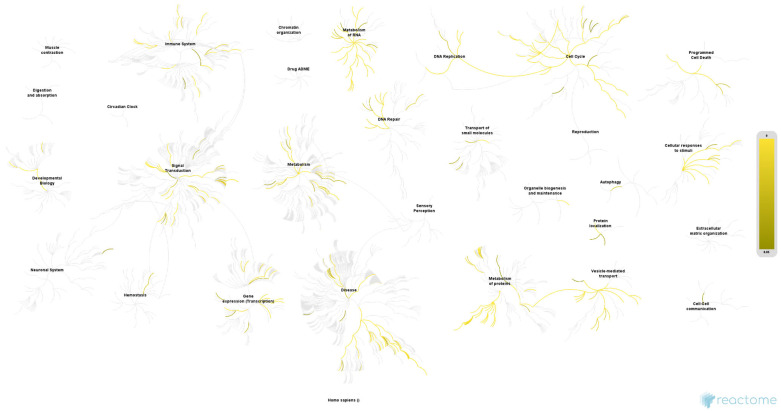
Proteome-wide overview of all Reactome pathways, organised hierarchically and showing sub-pathways. The graphical representation of the functional categories as nodes that ‘burst’ and spread out, establishing relationships with other pathways, serves to highlight the importance of metabolic regulatory processes, including the RNA activity, protein synthesis and regulation, cell cycle modulation, as well as signal transduction, in our results. The colour code indicates a gradient of over-representation, with yellow representing the most representative at the overexpression end and blue representing the most representative at the underexpression end. The light grey colour indicates that the pathways in question are not significantly over-represented. FDR < 1 × 10^−10^; entities ratio < 0.05; ratio found/list > 0.50.

**Figure 4 ijms-26-01559-f004:**
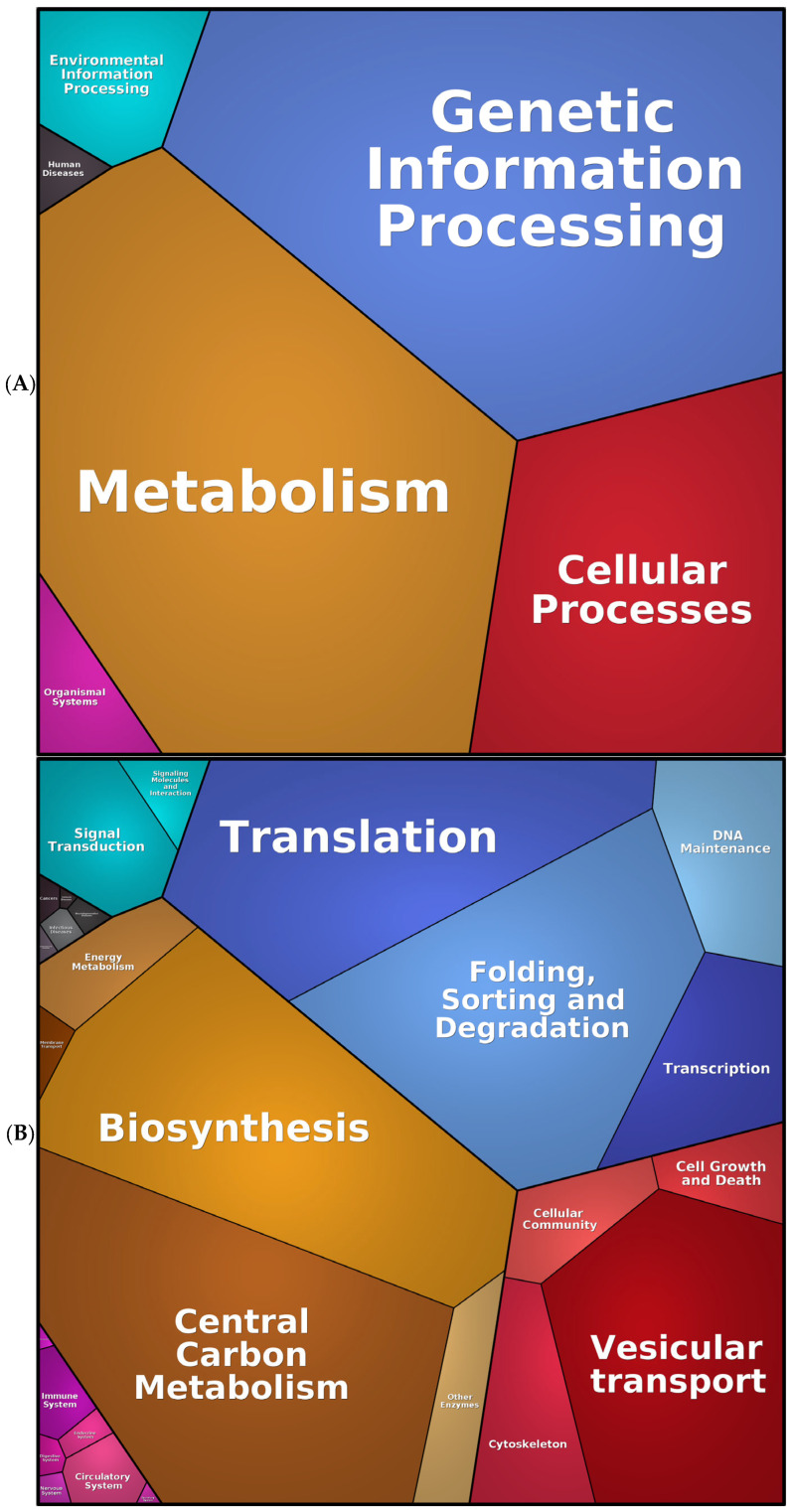
Proteomaps of the SK-BR-3 and SK-BR-3.rTP cells (from the list of 4239 proteins identified by the MS analysis). The three panels illustrate a progression from a global, core category level (**A**) to a more detailed (**B**), sub-process level of functional categorisation (**C**). Every tile (small polygon) represents one type of protein. Tiles are arranged and coloured according to the hierarchical KEGG pathway maps such that larger regions correspond to functional categories. The diagrams show three hierarchy levels. Tile sizes represent the mass fractions of proteins (protein abundances obtained by MS, multiplied by the protein chain lengths). Colour codes: blue, genetic information processing; brown, metabolism; red, cellular processes; green, signalling.

**Figure 5 ijms-26-01559-f005:**
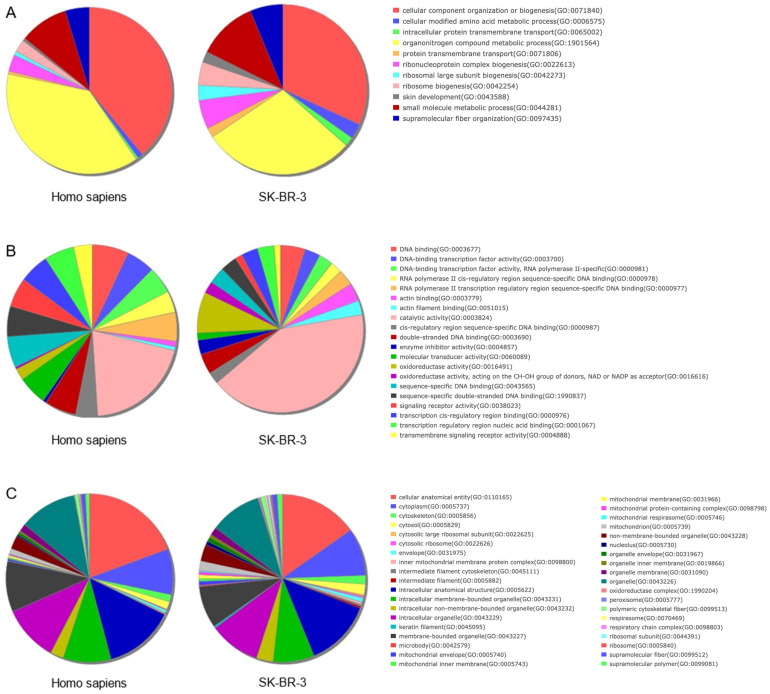
GO-based functional annotation of candidate proteins. The list of 618 DEPs was analysed by PANTHER for the following: (**A**) Biological process; (**B**) Molecular function; (**C**) Cellular component. FDR < 0.05; FC > 1.5; ratio found/list > 0.33.

**Figure 6 ijms-26-01559-f006:**
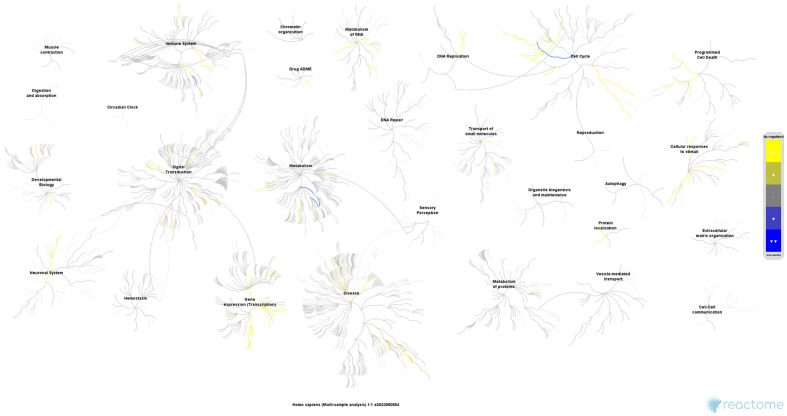
Proteome-wide overview of the results of the Reactome pathway analysis for the 618 DEPs. The analysis of the DEPs provided a more precise definition of the pathways suggested by the analysis of the total phenotype of the tumour line, emphasising the processes involved in cancer progression, the stress response, and the essential cellular metabolism. FDR < 1 × 10^−10^; entities ratio < 0.05; ratio found/list > 0.50.

**Figure 7 ijms-26-01559-f007:**
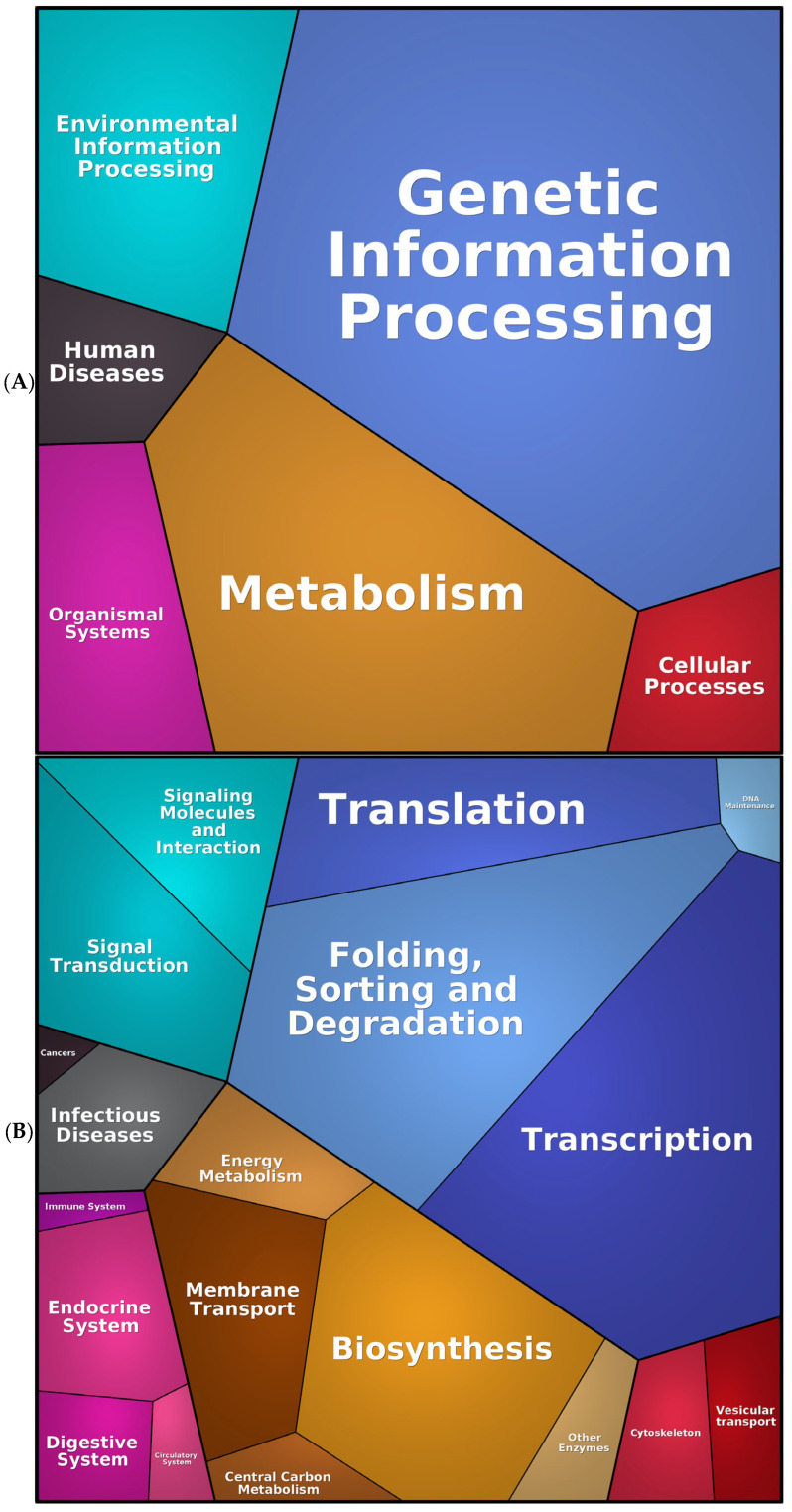
Proteomap from the list of 618 DEPs identified in the MS analysis of SK-BR-3 and SK-BR-3.rTP cells. The diagrams show three hierarchy levels: (**A**) Top functional categories according to Kegg Ontology databas; (**B**) Medium level of functional subcategories; (**C**) Deepest level.

**Figure 8 ijms-26-01559-f008:**
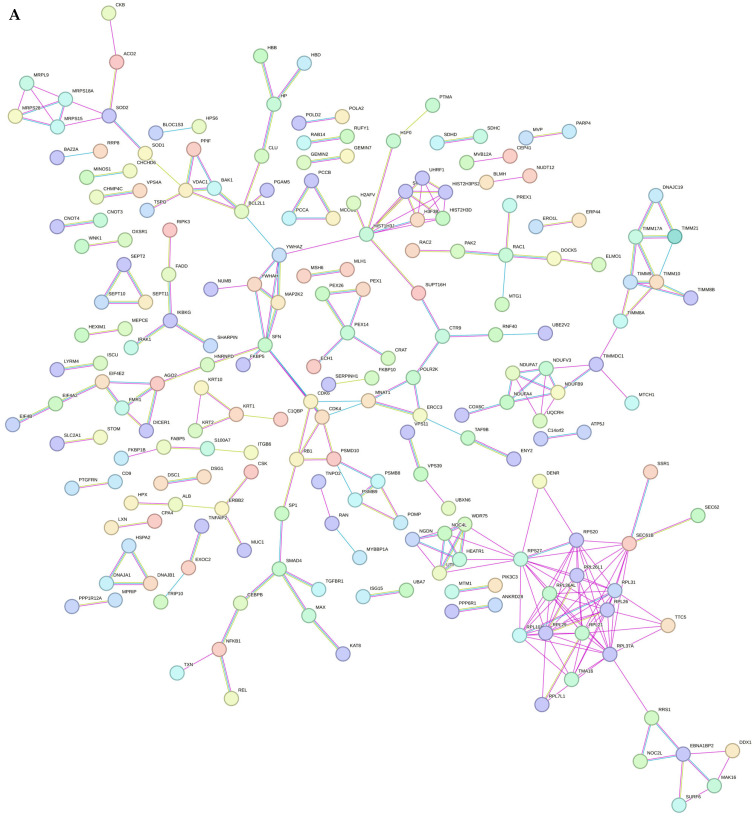
Protein interaction map produced through the STRING database of (**A**) The 618 DEPs. (**B**) The 55 proteins from the leading core. Each circle represents a protein, and the edges indicate both functional and physical protein associations. Only those proteins with interactions are shown in the networks. In panel (**B**), the blue halo surrounding a protein indicates its weight within the leading-edge subset. The intensity of the colour is indicative of the protein’s representativeness within the gene sets. The red filler colour indicates the presence of proteins that are regulated by microRNAs. *p*-value < 0.05; high confidence (>0.700).

**Figure 9 ijms-26-01559-f009:**
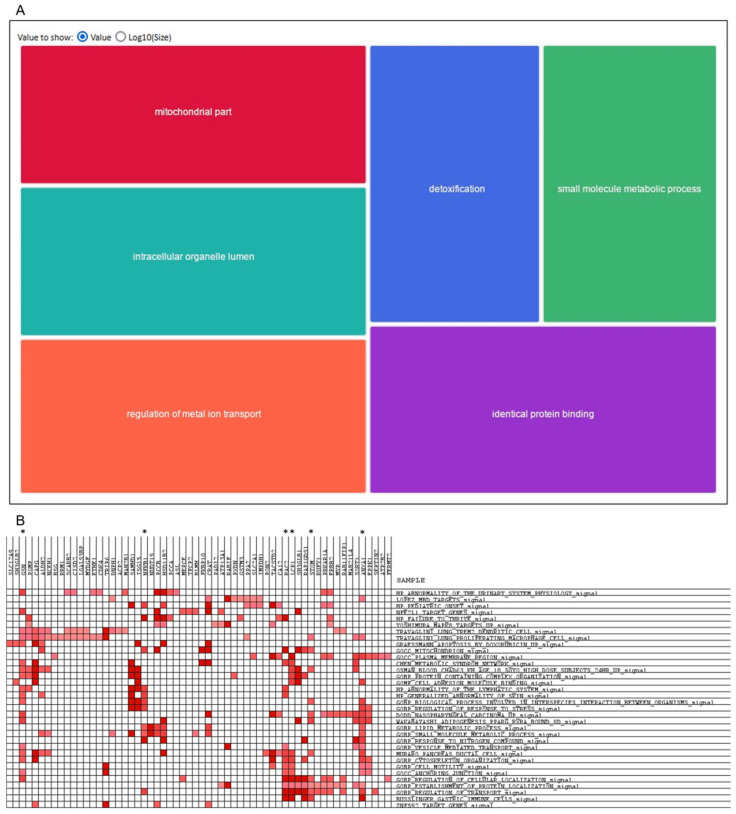
(**A**) A Revigo treemap representation providing a summary of the GO categories that were over-represented among the core DEPs of SK-BR-3 and SK-BR-3.rTP. A treemap was constructed for the 83 overexpressed and 118 downregulated proteins in the resistant strain. (**B**) The GSEA/LEA analysis identifying the core genes that underpin the enrichment signal of the gene set, as well as the genes that are shared across multiple gene sets. A heatmap providing a visual representation of the clustered genes within the leading-edge subsets, which are predominantly involved in cytoskeleton reorganisation, the reorganisation of cellular complexes, and metabolic processes. * denotes the most significant genes. (**C**) A STRING protein–protein interaction diagram of the core DEPs, indicating the presence of clusters with enhanced functional enrichment. The resulting clustering yielded sets of functionally associated proteins related to ribosome biogenesis (highlighted in lime green), the response to toxic substances (red), oxidoreductase activity (blue), and the mitochondrial matrix (yellow). FDR < 0.01; strength > 0.8; gene count ratio > 0.02.

**Figure 10 ijms-26-01559-f010:**
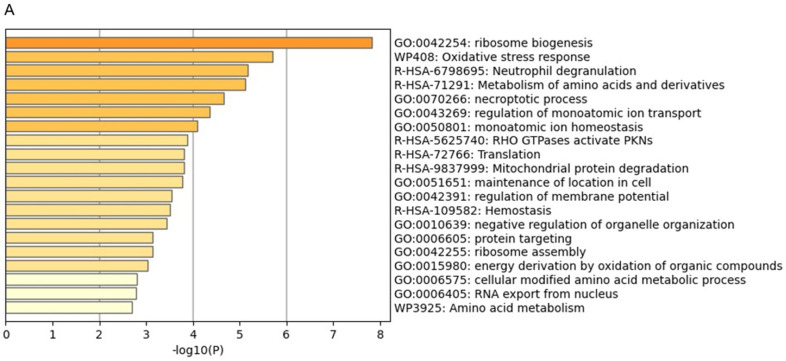
Functional enrichment analysis by Metascape. (**A**) Bar chart of clustered enrichment ontology categories (GO and KEGG terms) in the set of 83 proteins with increased abundance. Log10(q) < −3. (**B**) The protein–protein interaction network revealing the existence of three distinct clusters. A clustering algorithm was employed to identify densely connected protein neighbourhoods within the network. Subsequently, a GO enrichment analysis was conducted on each network to elucidate the underlying biological implications of the network components. The top three most significant *p*-value terms were retained, namely, ribosome biogenesis (indicated in red), monoatomic ion homeostasis (blue), and neutrophil extracellular trap formation (green). Log10(q) < −5. (**C**) Bar chart depicting the ontology terms that were found to be enriched in the set of 118 proteins with decreased abundance. Log10(q) < −3. (**D**) Protein–protein interaction network depicting the following three most significant groups: regulation of proteins (in red) and metabolic processes (blue and green). Log10(q) < −5.

**Figure 11 ijms-26-01559-f011:**
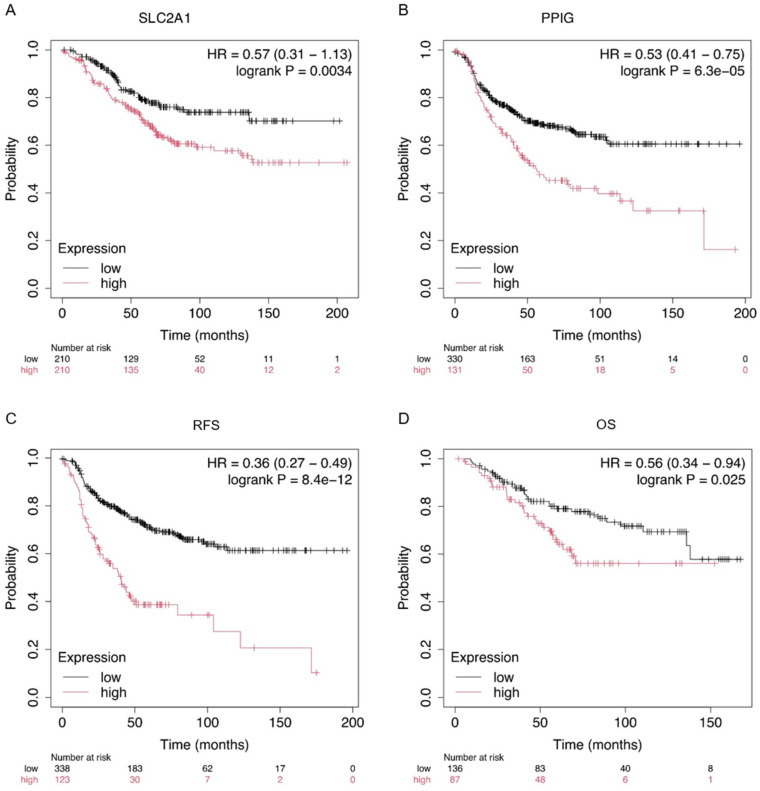
Associations between selected proteins, signatures, and clinical outcomes. Kaplan–Meier curves illustrate the correlations between protein expression levels and survival in HER2-positive breast cancer. (**A**) RFS for SLC2A1, a protein crucial in the glucose metabolism, highlighting its role in driving enriched gene sets. (**B**) RFS for PPIG, a protein in the leading-edge group and the 6-molecule signature, involved in protein biosynthesis. (**C**) Clinical significance of the 6-DEP survival signature in terms of RFS. (**D**) OS for the 6-DEP survival signature.

**Figure 12 ijms-26-01559-f012:**
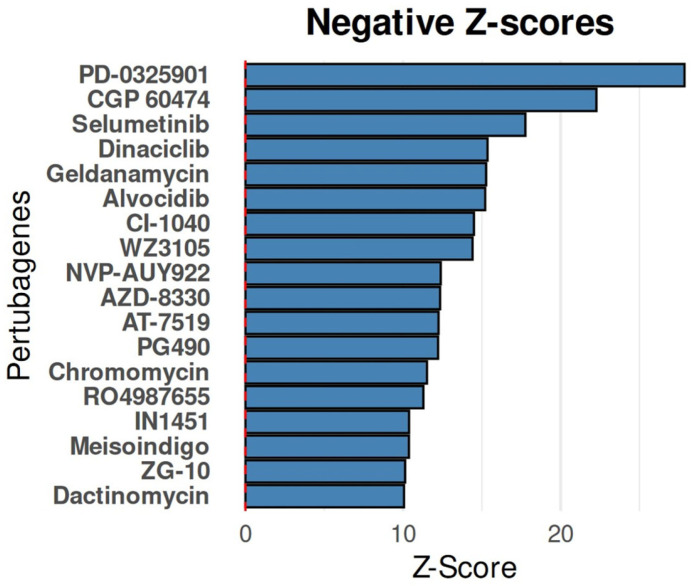
Connectivity map analysis of DEP signatures. Top-ranked perturbagens predicted by the iLINCS CMap analysis of the resistance signature with negative correlation. These results indicate chemicals reversing the resistance signatures.

**Table 1 ijms-26-01559-t001:** List of the top statistically over- and under-represented functional ontology categories in the SK-BR-3 phenotype, both in the parental state and in the resistant state. The total group of 4239 identified proteins were classified in PANTHER based on their functions using GO terms, which include the following: (**A**). biological process; (**B**). the molecular function; (**C**). the cellular component.

A. PANTHER GO-Slim Biological Process	Homo s.	SK-BR-3	Enrichment	*p*-Value	FDR	GO ID
Ribosomal subunit export from nucleus	11	11	4.78	4.40 × 10^−4^	2.67 × 10^−3^	GO:0000054
Mitochondrial translation	14	13	4.44	2.16 × 10^−4^	1.42 × 10^−3^	GO:0032543
COPII-coated vesicle budding	12	11	4.39	7.12 × 10^−4^	4.07 × 10^−3^	GO:0090114
Cytoplasmic translational initiation	14	12	4.10	6.13 × 10^−4^	3.58 × 10^−3^	GO:0002183
Inner mitochondrial membrane organization	13	11	4.05	1.11 × 10^−3^	5.98 × 10^−3^	GO:0007007
Cleavage involved in rRNA processing	17	14	3.94	2.88 × 10^−4^	1.82 × 10^−3^	GO:0000469
Maturation of LSU-rRNA from tricistronic rRNA transcript	11	9	3.91	3.67 × 10^−3^	1.62 × 10^−2^	GO:0000463
Ribosomal large subunit biogenesis	52	42	3.86	5.85 × 10^−10^	1.14 × 10^−8^	GO:0042273
Spliceosomal snRNP assembly	15	12	3.83	9.31 × 10^−4^	5.13 × 10^−3^	GO:0000387
Branched-chain amino acid metabolic process	10	8	3.83	6.70 × 10^−3^	2.67 × 10^−2^	GO:0009081
Endonucleolytic cleavage of tricistronic rRNA transcript	10	8	3.83	6.70 × 10^−3^	2.66 × 10^−2^	GO:0000479
Endonucleolytic cleavage involved in rRNA processing	10	8	3.83	6.70 × 10^−3^	2.66 × 10^−2^	GO:0000478
Protein transport to vacuole involved in ubiquitin-dependent	10	8	3.83	6.70 × 10^−3^	2.65 × 10^−2^	GO:0043328
ATP synthesis coupled electron transport	23	18	3.74	6.39 × 10^−5^	4.84 × 10^−4^	GO:0042773
Mitochondrial ATP synthesis coupled electron transport	22	17	3.70	1.13 × 10^−4^	8.09 × 10^−4^	GO:0042775
**B. PANTHER GO-Slim Molecular Function**	**Homo s.**	**SK-BR-3**	**Enrichment**	***p*-Value**	**FDR**	**GO ID**
Structural constituent of ribosome	111	100	4.31	4.23 × 10^−24^	1.22 × 10^−22^	GO:0003735
rRNA binding	26	23	4.23	1.54 × 10^−6^	1.28 × 10^−5^	GO:0019843
Proteasome binding	8	7	4.19	8.10 × 10^−3^	2.99 × 10^−2^	GO:0070628
Translation initiation factor activity	34	26	3.66	2.18 × 10^−6^	1.70 × 10^−5^	GO:0003743
Structural constituent of nuclear pore	19	14	3.52	6.31 × 10^−4^	3.22 × 10^−3^	GO:0017056
Translation elongation factor activity	15	11	3.51	2.45 × 10^−3^	1.11 × 10^−2^	GO:0003746
snoRNA binding	17	12	3.38	1.98 × 10^−3^	9.16 × 10^−3^	GO:0030515
Translation factor activity, RNA binding	56	39	3.33	5.65 × 10^−8^	6.17 × 10^−7^	GO:0008135
RNA polymerase II complex binding	16	11	3.29	3.50 × 10^−3^	1.46 × 10^−2^	GO:0000993
Translation regulator activity, nucleic acid binding	60	41	3.27	3.27 × 10^−8^	3.89 × 10^−7^	GO:0090079
Intramolecular transferase activity	21	14	3.19	1.27 × 10^−3^	6.18 × 10^−3^	GO:0016866
Basal RNA polymerase II transcription machinery binding	18	12	3.19	2.78 × 10^−3^	1.22 × 10^−2^	GO:0001099
Basal transcription machinery binding	18	12	3.19	2.78 × 10^−3^	1.21 × 10^−2^	GO:0001098
Ubiquitin binding	33	22	3.19	8.72 × 10^−5^	5.23 × 10^−4^	GO:0043130
ATP binding	41	27	3.15	1.28 × 10^−5^	8.60 × 10^−5^	GO:0005524
**C. PANTHER GO-Slim Cellular Component**	**Homo s.**	**SK-BR-3**	**Enrichment**	***p*-Value**	**FDR**	**GO ID**
Mitochondrial small ribosomal subunit	10	10	4.78	8.05 × 10^−4^	2.82 × 10^−3^	GO:0005763
Proteasome accessory complex	16	16	4.78	2.24 × 10^−5^	1.06 × 10^−4^	GO:0022624
Proteasome regulatory particle	16	16	4.78	2.24 × 10^−5^	1.05 × 10^−4^	GO:0005838
COP9 signalosome	8	8	4.78	2.73 × 10^−3^	8.50 × 10^−3^	GO:0008180
Golgi transport complex	6	6	4.78	9.51 × 10^−3^	2.58 × 10^−2^	GO:0017119
Endopeptidase complex	41	38	4.43	2.73 × 10^−10^	2.68 × 10^−9^	GO:1905369
Eukaryotic translation initiation factor 3 complex	13	12	4.42	3.91 × 10^−4^	1.49 × 10^−3^	GO:0005852
Proteasome complex	36	33	4.39	4.83 × 10^−9^	3.80 × 10^−8^	GO:0000502
COPI-coated vesicle	11	10	4.35	1.30 × 10^−3^	4.40 × 10^−3^	GO:0030137
Small ribosomal subunit	49	43	4.20	6.09 × 10^−11^	6.35 × 10^−10^	GO:0015935
Organellar ribosome	35	30	4.10	6.61 × 10^−8^	4.76 × 10^−7^	GO:0000313
Cytosolic small ribosomal subunit	35	30	4.10	6.61 × 10^−8^	4.69 × 10^−7^	GO:0022627
Mitochondrial ribosome	35	30	4.10	6.61 × 10^−8^	4.63 × 10^−7^	GO:0005761
Ribosome	126	107	4.06	2.67 × 10^−24^	6.82 × 10^−23^	GO:0005840
Ribosomal subunit	117	98	4.01	4.44 × 10^−22^	9.45 × 10^−21^	GO:0044391

**Table 2 ijms-26-01559-t002:** List of the most significant pathways and protein classes for the total proteins identified in the SK-BR-3, parental, and resistant samples.

PANTHER Pathways	Homo s.	SK-BR-3	Enrichment	*p*-Value	FDR	GO ID
TCA cycle	11	9	3.91	3.67 × 10^−3^	4.19 × 10^−2^	P00051
Ubiquitin proteasome pathway	58	41	3.38	2.08 × 10^−8^	1.67 × 10^−6^	P00060
Glycolysis	20	14	3.35	9.03 × 10^−4^	1.31 × 10^−2^	P00024
FAS signaling pathway	34	21	2.95	2.16 × 10^−4^	4.93 × 10^−3^	P00020
De novo purine biosynthesis	31	19	2.93	4.75 × 10^−4^	9.50 × 10^−3^	P02738
DNA replication	29	16	2.64	2.72 × 10^−3^	3.35 × 10^−2^	P00017
Parkinson disease	101	51	2.42	1.31 × 10^−6^	6.98 × 10^−5^	P00049
Ras Pathway	76	36	2.27	1.30 × 10^−4^	3.47 × 10^−3^	P04393
Cytoskeletal regulation by Rho GTPase	86	40	2.22	7.39 × 10^−5^	2.36 × 10^−3^	P00016
Huntington disease	152	63	1.98	1.15 × 10^−5^	4.61 × 10^−4^	P00029
EGF receptor signaling pathway	142	52	1.75	8.05 × 10^−4^	1.43 × 10^−2^	P00018
FGF signaling pathway	127	46	1.73	2.29 × 10^−3^	3.06 × 10^−2^	P00021
Integrin signalling pathway	200	68	1.63	8.17 × 10^−4^	1.31 × 10^−2^	P00034
CCKR signaling map	173	57	1.58	3.73 × 10^−3^	3.98 × 10^−2^	P06959
**PANTHER Protein Class**	**Homo s.**	**SK-BR-3**	**Enrichment**	***p*-Value**	**FDR**	**GO ID**
Translation elongation factor	15	14	4.46	1.19 × 10^−4^	5.19 × 10^−4^	PC00222
Ribosomal protein	177	151	4.08	6.62 × 10^−34^	1.85 × 10^−32^	PC00202
Aminoacyl-tRNA synthetase	41	34	3.97	1.58 × 10^−8^	1.07 × 10^−7^	PC00047
Translational protein	334	265	3.80	1.19 × 10^−54^	1.16 × 10^−52^	PC00263
Chaperonin	13	10	3.68	3.02 × 10^−3^	9.70 × 10^−3^	PC00073
Vesicle coat protein	45	34	3.61	7.71 × 10^−8^	4.88 × 10^−7^	PC00235
HSP70 family chaperone	15	11	3.51	2.45 × 10^−3^	8.44 × 10^−3^	PC00027
Translation factor	113	78	3.30	1.97 × 10^−14^	2.15 × 10^−13^	PC00223
Translation initiation factor	80	52	3.11	1.82 × 10^−9^	1.32 × 10^−8^	PC00224
Intermediate filament binding protein	14	9	3.08	1.09 × 10^−2^	3.19 × 10^−2^	PC00130
RNA splicing factor	135	85	3.01	5.01 × 10^−14^	5.17 × 10^−13^	PC00148
Isomerase	40	25	2.99	4.53 × 10^−5^	2.12 × 10^−4^	PC00135
ATP synthase	48	28	2.79	4.56 × 10^−5^	2.08 × 10^−4^	PC00002
Ligase	54	31	2.75	1.99 × 10^−5^	9.77 × 10^−5^	PC00142
RNA helicase	72	40	2.66	3.53 × 10^−6^	1.92 × 10^−5^	PC00032

**Table 3 ijms-26-01559-t003:** Reactome analysis of the quantified protein-identified pathways that were significantly enriched in the SK-BR-3 datasets. The table lists the 25 top pathways (both overexpressed and downregulated) with the *p*-value and the number of proteins that were identified in each pathway. It also includes the processes related to ribosomal regulation and mitochondrial functioning, as well as the mechanisms frequently altered in oncogenic transformation.

Pathway Name	Pathway Identifier	Direction	*p*-Value	FDR	No. Proteins	FC	Top-Regulated Pathways
Apoptosis	R-HSA-109581	Up	5.26 × 10^−4^	5.26 × 10^−4^	105	0.29	1
Activation and oligomerization of BAK protein	R-HSA-111452	Up	5.26 × 10^−4^	5.26 × 10^−4^	2	7.54	2
Neurotransmitter release cycle	R-HSA-112310	Up	5.26 × 10^−4^	5.26 × 10^−4^	9	3.10	3
Transmission across chemical synapses	R-HSA-112315	Up	5.26 × 10^−4^	5.26 × 10^−4^	60	0.75	4
Neuronal system	R-HSA-112316	Up	5.26 × 10^−4^	5.26 × 10^−4^	70	0.67	5
Formation of RNA pol II elongation complex	R-HSA-112382	Up	5.26 × 10^−4^	5.26 × 10^−4^	38	0.71	6
Activation of NF-kappaB in B cells	R-HSA-1169091	Up	5.26 × 10^−4^	5.26 × 10^−4^	50	0.20	7
Signaling by FGFR in disease	R-HSA-1226099	Up	5.26 × 10^−4^	5.26 × 10^−4^	21	0.36	8
Mitochondrial protein import	R-HSA-1268020	Up	5.26 × 10^−4^	5.26 × 10^−4^	46	2.04	9
Mitochondrial iron-sulfur cluster biogenesis	R-HSA-1362409	Up	5.26 × 10^−4^	5.26 × 10^−4^	6	3.00	10
RNA Pol II CTD phosphorylation interaction with CE	R-HSA-77069	Up	5.26 × 10^−4^	5.26 × 10^−4^	18	1.74	11
NOD1/2 signaling pathway	R-HSA-168638	Down	5.26 × 10^−4^	5.26 × 10^−4^	10	−0.59	12
Nucleotide-binding domain, NLR signaling pathways	R-HSA-168643	Down	5.26 × 10^−4^	5.26 × 10^−4^	18	−0.86	13
FGFR2 mutant receptor activation	R-HSA-1839126	Up	5.26 × 10^−4^	5.26 × 10^−4^	11	0.91	14
Signaling by FGFR	R-HSA-190236	Up	5.26 × 10^−4^	5.26 × 10^−4^	30	0.25	15
Estrogen biosynthesis	R-HSA-193144	Up	5.26 × 10^−4^	5.26 × 10^−4^	1	12.45	16
p75NTR signals via NF-kB	R-HSA-193648	Up	5.26 × 10^−4^	5.26 × 10^−4^	7	1.01	17
Degradation of beta-catenin by the destruction complex	R-HSA-195253	Up	5.26 × 10^−4^	5.26 × 10^−4^	57	0.13	18
Metabolism of steroid hormones	R-HSA-196071	Up	5.26 × 10^−4^	5.26 × 10^−4^	8	1.19	19
MicroRNA (miRNA) biogenesis	R-HSA-203927	Up	5.26 × 10^−4^	5.26 × 10^−4^	15	1.79	20
NF-kB is activated and signals survival	R-HSA-2095650	Up	5.26 × 10^−4^	5.26 × 10^−4^	5	1.82	21
Glutamate neurotransmitter release cycle	R-HSA-210500	Up	5.26 × 10^−4^	5.26 × 10^−4^	5	2.75	22
Synthesis of leukotrienes (LT) and eoxins (EX)	R-HSA-2142753	Down	5.26 × 10^−4^	5.26 × 10^−4^	3	−4.99	23
Synthesis of lipoxins (LX)	R-HSA-2142700	Down	5.26 × 10^−4^	5.26 × 10^−4^	1	−14.92	24
Arachidonic acid metabolism	R-HSA-2142753	Down	5.26 × 10^−4^	5.26 × 10^−4^	12	−0.67	25

**Table 4 ijms-26-01559-t004:** List of the 15 top ontological terms for each category.

GO Biological Process	Homo s.	618 Proteins	Enrichment	*p*-Value	FDR	GO ID
Maturation of LSU-rRNA	28	6	7.05	4.28 × 10^−4^	4.63 × 10^−2^	GO:0000470
Purine nucleotide catabolic process	48	10	6.85	6.78 × 10^−6^	1.71 × 10^−3^	GO:0006195
Purine-containing compound catabolic process	54	10	6.09	1.68 × 10^−5^	3.71 × 10^−3^	GO:0072523
Ribosomal large subunit biogenesis	76	14	6.06	3.76 × 10^−7^	1.84 × 10^−4^	GO:0042273
Protein transmembrane transport	64	11	5.65	1.19 × 10^−5^	2.79 × 10^−3^	GO:0071806
Intracellular protein transmembrane transport	53	9	5.59	8.04 × 10^−5^	1.30 × 10^−2^	GO:0065002
Fatty acid catabolic process	85	14	5.42	1.24 × 10^−6^	4.41 × 10^−4^	GO:0009062
Cellular oxidant detoxification	93	15	5.30	6.57 × 10^−7^	2.78 × 10^−4^	GO:0098869
Glutathione metabolic process	57	9	5.19	1.32 × 10^−4^	1.83 × 10^−2^	GO:0006749
Cellular detoxification	108	17	5.18	1.63 × 10^−7^	9.45 × 10^−5^	GO:1990748
Monocarboxylic acid catabolic process	110	17	5.08	2.06 × 10^−7^	1.11 × 10^−4^	GO:0072329
Fatty acid beta-oxidation	60	9	4.93	1.87 × 10^−4^	2.47 × 10^−2^	GO:0006635
Detoxification	136	20	4.84	3.68 × 10^−8^	2.88 × 10^−5^	GO:0098754
Cellular response to toxic substance	116	17	4.82	4.03 × 10^−7^	1.92 × 10^−4^	GO:0097237
Keratinization	83	12	4.76	2.30 × 10^−5^	4.68 × 10^−3^	GO:0031424
**GO Molecular Function**	**Homo s.**	**618 Proteins**	**Enrichment**	***p*-Value**	**FDR**	**GO ID**
Antioxidant activity	88	15	5.61	3.51 × 10^−7^	1.25 × 10^−4^	GO:0016209
Chaperone binding	107	16	4.92	6.93 × 10^−7^	1.92 × 10^−4^	GO:0051087
Oxidoreductase activity, NAD or NADP as acceptor	128	19	4.88	7.06 × 10^−8^	3.19 × 10^−5^	GO:0016616
Oxidoreductase activity, acting on the CH-CH group of donors	61	9	4.85	2.09 × 10^−4^	2.54 × 10^−2^	GO:0016627
Oxidoreductase activity, acting on CH-OH group of donors	140	19	4.46	2.48 × 10^−7^	9.49 × 10^−5^	GO:0016614
Structural constituent of ribosome	168	17	3.33	3.64 × 10^−5^	6.03 × 10^−3^	GO:0003735
Oxidoreductase activity	741	64	2.84	5.78 × 10^−13^	4.80 × 10^−10^	GO:0016491
Cadherin binding	325	24	2.43	1.47 × 10^−4^	1.97 × 10^−2^	GO:0045296
Cell adhesion molecule binding	552	33	1.97	4.26 × 10^−4^	4.82 × 10^−2^	GO:0050839
RNA binding	1666	99	1.95	4.01 × 10^−10^	2.49 × 10^−7^	GO:0003723
Protein homodimerization activity	712	40	1.85	3.78 × 10^−4^	4.37 × 10^−2^	GO:0042803
Identical protein binding	2146	119	1.82	2.85 × 10^−10^	2.03 × 10^−7^	GO:0042802
Protein-containing complex binding	1312	66	1.65	9.87 × 10^−5^	1.40 × 10^−2^	GO:0044877
Small molecule binding	2515	125	1.63	7.05 × 10^−8^	3.51 × 10^−5^	GO:0036094
Nucleotide binding	2170	107	1.62	1.04 × 10^−6^	2.60 × 10^−4^	GO:0000166
**GO Cellular Component**	**Homo s.**	**618 Proteins**	**Enrichment**	***p*-Value**	**FDR**	**GO ID**
Mitochondrial intermembrane space protein transporter complex	6	4	21.93	1.37 × 10^−4^	4.74 × 10^−3^	GO:0042719
IgG immunoglobulin complex	10	4	13.16	5.94 × 10^−4^	1.71 × 10^−2^	GO:0071735
Cornified envelope	59	12	6.69	1.02 × 10^−6^	5.36 × 10^−5^	GO:0001533
Blood microparticle	144	25	5.71	3.23 × 10^−11^	3.48 × 10^−9^	GO:0072562
Cytosolic large ribosomal subunit	59	9	5.02	1.67 × 10^−4^	5.50 × 10^−3^	GO:0022625
Azurophil granule lumen	90	13	4.75	1.07 × 10^−5^	4.63 × 10^−4^	GO:0035578
Mitochondrial intermembrane space	84	12	4.70	2.55 × 10^−5^	1.02 × 10^−3^	GO:0005758
Organelle envelope lumen	94	12	4.20	6.89 × 10^−5^	2.47 × 10^−3^	GO:0031970
Cytosolic ribosome	105	12	3.76	1.79 × 10^−4^	5.80 × 10^−3^	GO:0022626
Actin filament bundle	80	9	3.70	1.25 × 10^−3^	3.19 × 10^−2^	GO:0032432
Azurophil granule	154	17	3.63	1.31 × 10^−5^	5.58 × 10^−4^	GO:0042582
Primary lysosome	154	17	3.63	1.31 × 10^−5^	5.46 × 10^−4^	GO:0005766
Oxidoreductase complex	125	13	3.42	2.30 × 10^−4^	7.22 × 10^−3^	GO:1990204
Inner mitochondrial membrane protein complex	156	16	3.37	5.28 × 10^−5^	1.96 × 10^−3^	GO:0098800
Secretory granule lumen	320	32	3.29	1.98 × 10^−8^	1.35 × 10^−6^	GO:0034774

**Table 5 ijms-26-01559-t005:** List of pathways found in the Reactome analysis for the 618 DEPs. The terms reflect the molecular mechanisms related to the regulation of the cell cycle, gene and protein regulation, metabolic pathways, mitochondrial function, apoptosis, and the evasion of senescence. For reasons of space, only the first 25 pathways are listed.

Pathway Name	Pathway Identifier	Direction	*p*-Value	FDR	No. Proteins	FC	Top-Regulated Pathways
Displacement of DNA glycosylase by APEX1	R-HSA-110357	Up	5.26 × 10^−4^	5.26 × 10^−4^	2	5.78	1
Activation of NOXA and translocation to mitochondria	R-HSA-111448	Up	5.26 × 10^−4^	5.26 × 10^−4^	13	3.56	2
Activation and oligomerization of BAK protein	R-HSA-111452	Up	5.26 × 10^−4^	5.26 × 10^−4^	6	2.93	3
BH3-only proteins associate with and inactivate anti-apoptotic BCL-2 members	R-HSA-111453	Up	5.26 × 10^−4^	5.26 × 10^−4^	15	2.36	4
Neurotransmitter release cycle	R-HSA-112310	Up	5.26 × 10^−4^	5.26 × 10^−4^	12	3.95	5
Activation, translocation and oligomerization of BAX	R-HSA-114294	Up	5.26 × 10^−4^	5.26 × 10^−4^	3	5.16	6
Activation of BH3-only proteins	R-HSA-114452	Up	5.26 × 10^−4^	5.26 × 10^−4^	40	2.14	7
Activation of NF-kappaB in B cells	R-HSA-1169091	Up	5.26 × 10^−4^	5.26 × 10^−4^	30	1.77	8
Signaling by FGFR in disease	R-HSA-1226099	Up	5.26 × 10^−4^	5.26 × 10^−4^	23	1.74	9
Regulation of gene expression by Hypoxia-inducible Factor	R-HSA-1234158	Up	5.26 × 10^−4^	5.26 × 10^−4^	3	9.17	10
Mitochondrial protein import	R-HSA-1268020	Up	5.26 × 10^−4^	5.26 × 10^−4^	23	3.81	11
Adaptive immune system	R-HSA-1280218	Up	5.26 × 10^−4^	5.26 × 10^−4^	164	1.64	12
Activation of PUMA and translocation to mitochondria	R-HSA-139915	Up	5.26 × 10^−4^	5.26 × 10^−4^	14	3.32	13
Inhibition of the proteolytic activity of APC/C for the onset of anaphase	R-HSA-141405	Down	5.26 × 10^−4^	5.26 × 10^−4^	5	−4.46	14
Inactivation of APC/C via direct inhibition of the APC/C complex	R-HSA-141430	Down	5.26 × 10^−4^	5.26 × 10^−4^	5	−4.46	15
Methylation	R-HSA-5334118	Up	5.26 × 10^−4^	5.26 × 10^−4^	2	6.03	16
MyD88:MAL(TIRAP) cascade initiated on plasma membrane	R-HSA-166058	Up	5.26 × 10^−4^	5.26 × 10^−4^	92	1.95	17
Toll Like Receptor TLR1:TLR2 cascade	R-HSA-168179	Up	5.26 × 10^−4^	5.26 × 10^−4^	92	1.95	18
Toll Like Receptor TLR6:TLR2 cascade	R-HSA-168188	Up	5.26 × 10^−4^	5.26 × 10^−4^	92	1.95	19
DDX58/IFIH1-mediated induction of interferon-alpha/beta	R-HSA-168928	Up	5.26 × 10^−4^	5.26 × 10^−4^	80	1.31	20
Conversion from APC/C:Cdc20 to APC/C:Cdh1 in late anaphase	R-HSA-176407	Down	5.26 × 10^−4^	5.26 × 10^−4^	4	−1.67	21
Toll Like Receptor 2 (TLR2) cascade	R-HSA-168898	Up	5.26 × 10^−4^	5.26 × 10^−4^	92	1.95	22
Signalling to STAT3	R-HSA-198745	Up	5.26 × 10^−4^	5.26 × 10^−4^	4	3.04	23
TCR signaling	R-HSA-202403	Up	5.26 × 10^−4^	5.26 × 10^−4^	44	1.20	24
MicroRNA (miRNA) biogenesis	R-HSA-203927	Up	5.26 × 10^−4^	5.26 × 10^−4^	12	2.34	25

**Table 6 ijms-26-01559-t006:** GSEA of the differentially regulated set of genes in SK-BR3.rTP1 cells as compared to SK-BR3 (Human MSigDB Collection C6). The twenty top gene sets are shown. NES: Normalised Enrichment Score.

Gene Set Name	Proteins from Study Found in Pathway	NES
RB_P107_DN.V1_UP	FKBP10, RRM1, ETHE1, LGALS3BP, SNCG, S100A4	1.80
E2F1_UP.V1_DN	NFKB1, ANXA1, STOM, LXN, NOTCH2, USP19, ASL, STAT6	1.69
CYCLIN_D1_KE_.V1_DN	CAPG, HRG, ERBB2, TFCP2, BAG1, KRT13, PPIG, KRT17	1.65
HOXA9_DN.V1_UP	SAMHD1, ISG15, CARD9, SLC30A1, FAM98A, CYP1B1, DICER1	1.58
CYCLIN_D1_UP.V1_DN	CAPG, HRG, TFCP2, ACAA1, BAG1, ITGB6, S100A4, PPIG	1.56
NRL_DN.V1_DN	NFKB1, ALDH2, EPM2AIP1, RIOX2, TNPO2, SUPT16H	1.48
E2F1_UP.V1_UP	CLN6, POLA2, CRYZ, RRM1, UBR2, RB1, AGO2, TFAM	1.43
KRAS.KIDNEY_UP.V1_DN	HRG, CARD9, ACE, KRT17, KRT2	1.43
RB_P130_DN.V1_UP	HINT1, CHCHD6, RIPK3, POLD2, ICA1	1.43
STK33_SKM_UP	NFXL1, ANXA1, LGALS1, TNFAIP2, CD9, APPL2	1.34
MEK_UP.V1_DN	SAMHD1, ISG15, SEC24D, ANXA9, PRPF38B, H1-2	1.26
GCNP_SHH_UP_EARLY.V1_DN	LCP1, CRABP2, MAN2B1, NUB1, KIAA1217	1.25
ESC_V6.5_UP_EARLY.V1_DN	GSTP1, ANXA1, GSN, S100A4, FHL2, SFN	1.25
STK33_UP	NFXL1, ANXA1, NCEH1, GAN, CD9, APPL2, SEPTIN10	1.23
RB_DN.V1_DN	MVP, PRKAR1A, RB1, LAMTOR3, MYORG, CD9	1.23
TBK1.DN.48HRS_UP	CDK4, UBQLN2, CNDP2, OXSR1, CNOT4	1.21
P53_DN.V1_UP	SH3GLB1, TACSTD2, ANXA1, MARK2, SLC16A2, AZGP1, MARCKS, UQCRH, CD9, TMEM30B, FHL2, SDC4, AGR2	1.19
GCNP_SHH_UP_LATE.V1_DN	SLC2A1, MAN2B1, PSMB8, IKBKG, MYORG, UBE2V2	1.19
BMI1_DN_MEL18_DN.V1_UP	LCP1, ITGB6, CPA4, ADGRE5, NT5E	1.17
AKT_UP_MTOR_DN.V1_UP	TACSTD2, PON2, MVP, SLC6A6, VAT1, PYGL	1.17

**Table 7 ijms-26-01559-t007:** List of proteins in the STRING network that matched each of the corresponding GO terms highlighted in Figure 9C.

#Color	Category	Term ID	Term Description	Observed Gene Count	Background Gene Count	FDR	Matching Proteins
limegreen	GO Process	GO:0042254	Ribosome biogenesis	10	299	0.0026	C1QBP, DDX18, RRS1, HEATR1, MYBBP1A, NGDN, EBNA1BP2, RPL7L1, RPL26L1, RPS27
red	GO Process	GO:0009636	Response to toxic substance	8	229	0.0098	NQO1, AKR7A3, GLRX, MGST1, TXNRD2, CPS1, SOD2, GSTZ1
blue	GO Function	GO:0016491	Oxidoreductase activity	14	731	0.0061	YWHAH, SORD, NDUFB9, TMX2, NQO1, VAT1, AKR7A3, GLRX, ERO1A, MGST1, TXNRD2, SOD2, GSTZ1, MTHFD1L
yellow	GO Component	GO:0005759	Mitochondrial matrix	14	494	8.08 × 10^−6^	ACO2, PPIF, C1QBP, MCCC1, MRPS28, ECI1, TSFM, GCSH, TXNRD2, CPS1, SOD2, GSTZ1, MTHFD1L, REXO2

## Data Availability

The MS proteomics data have been deposited to the ProteomeXchange Consortium via the PRIDE partner repository with the dataset identifier PXD045804.

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
