# Peer review of "Adaptive Proteomic Changes in Protein Metabolism and Mitochondrial Alterations Associated with Resistance to Trastuzumab and Pertuzumab Therapy in HER2-Positive Breast Cancer"

_ijms, 2025, doi:10.3390/ijms26041559_

Round 1
Reviewer 1 Report
Comments and Suggestions for Authors
The manuscript explores proteomic and mitochondrial alterations in HER2-positive breast cancer cell lines resistant to trastuzumab and pertuzumab therapy. Utilizing label-free mass spectrometry-based proteomics, it identifies differentially expressed proteins and bioinformatic analyses of pathways related to resistance mechanisms. The study highlights mitochondrial activity and detoxification processes as pivotal resistance mechanisms and proposes therapeutic targets. However, some points need to be addressed to accept this manuscript for publication.
Introduction
- Provide a more concise background on HER2-positive breast cancer and resistance mechanisms.
Methods
- The methods lack sufficient detail about how findings were validated experimentally.
- The bioinformatics approach for pathway analysis and protein interaction needs more elaboration.
Results
- The results section is exhaustive but difficult to follow. Please use subheadings to organize your “result section” better.
- Figures such as proteomaps and STRING networks lack clarity. Improve resolution and add more detailed legends.
Discussion
- Discussion is too long to follow. Please shorten “the discussion section” in 2-2.5 pages.
- The discussion fails to link findings to potential clinical applications. Expand on how the identified proteins can be targeted therapeutically.
- Acknowledge the study's limitations, particularly the in vitro nature of the models used.
Figures and Tables
- Please ensure that all figures have high resolution and are fully labeled.
- Revise all Figure and table captions to include descriptions of statistical analysis and key findings
Comments on the Quality of English Language
NA
Author Response
RESPONSE TO REVIEWER 1 COMMENTS
Review report form 1
The manuscript explores proteomic and mitochondrial alterations in HER2-positive breast cancer cell lines resistant to trastuzumab and pertuzumab therapy. Utilizing label-free mass spectrometry-based proteomics, it identifies differentially expressed proteins and bioinformatic analyses of pathways related to resistance mechanisms. The study highlights mitochondrial activity and detoxification processes as pivotal resistance mechanisms and proposes therapeutic targets. However, some points need to be addressed to accept this manuscript for publication.
We are extremely grateful for the reviewer's suggestions and will endeavour to address the following questions.
Introduction.
- Provide a more concise background on HER2-positive breast cancer and resistance mechanisms.
A revision of the Introduction has been conducted, with a focus on enhancing the text's clarity and conciseness to ensure optimal readability. Concurrently, the English language has been revised to eliminate minor language imperfections.
Methods.
- The methods lack sufficient detail about how findings were validated experimentally.
All experimental validations are described in great detail in the preceding article, whose reference can be found in reference #26:
- Sanz-Álvarez, M.; Luque, M.; Morales-Gallego, M.; Cristóbal, I.; Ramírez-Merino, N.; Rangel, Y.; Izarzugaza, Y.; Eroles, P.; Albanell, J.; Madoz-Gúrpide, J.; et al. Generation and Characterization of Trastuzumab/Pertuzumab-Resistant HER2-Positive Breast Cancer Cell Lines. Int. J. Mol. Sci. 2024, 25, 207, doi:10.3390/ijms25010207.
- The bioinformatics approach for pathway analysis and protein interaction needs more elaboration.
In response to the reviewer's request, the descriptions of the pathway enrichment and protein interaction analyses have been expanded. Moreover, we have incorporated statistical data pertaining to the outcomes of the aforementioned analyses within the figure captions.
Results.
- The results section is exhaustive but difficult to follow. Please use subheadings to organize your “result section” better.
It is acknowledged that the Results section is replete with information and can be challenging to interpret. To facilitate comprehension of the findings, the Results section has been organised into subsections, which are further divided into paragraphs that refer to different methodological approaches. The use of subheadings in these paragraphs has been avoided to minimise potential confusion.
- Figures such as Proteomaps and STRING networks lack clarity. Improve resolution and add more detailed legends.
The images that were initially produced were of a high resolution, and thus they were submitted to the journal via the manuscript submission system. However, the resolution of the version of the manuscript prepared by the journal's editorial team is lower than that provided by the authors. In some cases, this lower resolution is insufficient to appreciate the details of the analyses. This is an editorial problem in the preparation of the manuscript. We will resubmit the original files at full resolution; however, it is essential that the journal maintains this resolution to ensure the integrity of the visual content.
Discussion.
- Discussion is too long to follow. Please shorten “the discussion section” in 2-2.5 pages.
In response to the reviewer's recommendation, the length of the Discussion section has been reduced by three pages. In addition, we have introduced sub-sections in order to facilitate the organisation of the text and to enhance its readability.
- The discussion fails to link findings to potential clinical applications. Expand on how the identified proteins can be targeted therapeutically.
An attempt was made to demonstrate the translational significance of the study and to establish potential connections between the findings of our analysis and the possible clinical applications thereof. This is most evident in the final section of the Results, which corresponds to the Discussion section on the identification of therapeutic candidates for reversing resistance in HER2-positive breast cancer.
- Acknowledge the study's limitations, particularly the in vitro nature of the models used.
A paragraph has finally been appended to the end of the Discussion section, the purpose of which is to indicate the limitations of the study.
Figures and Tables.
- Please ensure that all figures have high resolution and are fully labeled.
As outlined above, please find my response to your query.
- Revise all Figure and table captions to include descriptions of statistical analysis and key findings.
All figure and table captions have undergone revision. A number of supplementary indications have been incorporated with a view to elucidating that which was previously delineated in the preceding version. It is evident that all figures and tables reference their respective statistical analyses, whether in the figure/table itself or in the reference table from which the corresponding diagram was constructed.
English language.
The manuscript has undergone a comprehensive revision and editing process to ensure its accuracy and clarity in the English language.

Reviewer 2 Report
Comments and Suggestions for Authors
Madoz- Gúrpide studied the HER2-positive breast cancer cells using parental SK-BR-3 cells and derived trastuzumab and pertuzumab resistant cells. They performed specific LC-MS/MS analysis on the proteins from the lysate of the two cells. They first compared the identified proteins with general human protein expression profiles. They successfully quantified over 4000 identified proteins and found 349 upregulated and 269 downregulated proteins in trastuzumab and pertuzumab resistant cells. They perform a lot of analysis on the identified proteins and differentially expressed proteins such as gene ontology analysis, Reactome pathway analysis, Proteomap analysis, STRING protein interaction network analysis, etc. They found significant protein expression differences, especially for proteins that are responsible for mitochondrial activity and detoxification processes. This is a nice report and will benefit the breast cancer research community. The manuscript may provide new insights into the resistance of HER2-positive breast cancer to trastuzumab and pertuzumab. The following issues should be resolved to improve the presentation.
Comments:
1. Section 3.1 and 3.2: The authors compared the proteins identified in SK-BR-3 cell lines (parental and resistant) with the general human proteome. I couldn’t find the “human proteome” used for analysis here. Please specify it here/in the method section to make things clear. In addition, proteins have tissue-specific expressions. It is better to compare the protein expression of the two breast cancer cell lines with that of normal breast tissue, such as the mammary gland from where the SK-BR-3 cell line was derived.
2. Most of the figures in the manuscript have very low resolution especially the Reactome pathways and Proteomaps. It is impossible to see what are in the figures. Please use high resolution images.
Author Response
RESPONSE TO REVIEWER 2 COMMENTS
Review report form 1
Madoz-Gúrpide studied the HER2-positive breast cancer cells using parental SK-BR-3 cells and derived trastuzumab and pertuzumab resistant cells. They performed specific LC-MS/MS analysis on the proteins from the lysate of the two cells. They first compared the identified proteins with general human protein expression profiles. They successfully quantified over 4000 identified proteins and found 349 upregulated and 269 downregulated proteins in trastuzumab and pertuzumab resistant cells. They perform a lot of analysis on the identified proteins and differentially expressed proteins such as gene ontology analysis, Reactome pathway analysis, Proteomap analysis, STRING protein interaction network analysis, etc. They found significant protein expression differences, especially for proteins that are responsible for mitochondrial activity and detoxification processes. This is a nice report and will benefit the breast cancer research community. The manuscript may provide new insights into the resistance of HER2-positive breast cancer to trastuzumab and pertuzumab. The following issues should be resolved to improve the presentation.
We are appreciative of the reviewer's contributions and will address the queries raised in the following paragraphs.
Comments and suggestions for authors.
- Section 3.1 and 3.2: The authors compared the proteins identified in SK-BR-3 cell lines (parental and resistant) with the general human proteome. I couldn’t find the “human proteome” used for analysis here. Please specify it here/in the method section to make things clear. In addition, proteins have tissue-specific expressions. It is better to compare the protein expression of the two breast cancer cell lines with that of normal breast tissue, such as the mammary gland from where the SK-BR-3 cell line was derived.
As suggested by the reviewer, reference to the 'human proteome' has now been included in the Methods section. In any case, for a comprehensive overview of the methodologies employed in the proteomic study that formed the basis of the present analysis, please refer to the article published by our group a few months ago (Ref.: 26).
- Sanz-Álvarez, M.; Luque, M.; Morales-Gallego, M.; Cristóbal, I.; Ramírez-Merino, N.; Rangel, Y.; Izarzugaza, Y.; Eroles, P.; Albanell, J.; Madoz-Gúrpide, J.; et al. Generation and Characterization of Trastuzumab/Pertuzumab-Resistant HER2-Positive Breast Cancer Cell Lines. Int. J. Mol. Sci. 2024, 25, 207, doi:10.3390/ijms25010207.
In particular, MS/MS spectra acquired on the samples were analysed using Proteome Discoverer v2.5 software (Thermo Scientific) with the MASCOT v.2.8 search engine. The UniProt database with taxonomic restriction to humans was used (release 2023_03).
Furthermore, we have worked with breast tissue-specific protein/genes databases whenever possible.
- Most of the figures in the manuscript have very low resolution especially the Reactome pathways and Proteomaps. It is impossible to see what are in the figures. Please use high resolution images.
The images that were initially produced were of a high resolution, and thus they were submitted to the journal via the manuscript submission system. However, the resolution of the version of the manuscript prepared by the journal's editorial team is lower than that provided by the authors. In some cases, this lower resolution is insufficient to appreciate the details of the analyses. This is an editorial problem in the preparation of the manuscript. We will resubmit the original files at full resolution; however, it is essential that the journal maintains this resolution to ensure the integrity of the visual content.
English language.
The manuscript has undergone a comprehensive revision and editing process to ensure its accuracy and clarity in the English language.
